# Atomic structure of a mitochondrial complex I intermediate from vascular plants

**Maria Maldonado[1], Abhilash Padavannil[1], Long Zhou[1], Fei Guo[1,2], James A Letts[1]***

[1]Department of Molecular and Cellular Biology, University of California Davis, Davis, United States; [2]BIOEM Facility, University of California Davis, Davis, United States

**Abstract** Respiration, an essential metabolic process, provides cells with chemical energy. In eukaryotes, respiration occurs via the mitochondrial electron transport chain (mETC) composed of several large membrane-protein complexes. Complex I (CI) is the main entry point for electrons into the mETC. For plants, limited availability of mitochondrial material has curbed detailed biochemical and structural studies of their mETC. Here, we present the cryoEM structure of the known CI assembly intermediate CI* from *Vigna radiata* at 3.9 Å resolution. CI* contains CI's NADH-binding and CoQ-binding modules, the proximal-pumping module and the plant-specific γ-carbonic-anhydrase domain (γCA). Our structure reveals significant differences in core and accessory subunits of the plant complex compared to yeast, mammals and bacteria, as well as the details of the γCA domain subunit composition and membrane anchoring. The structure sheds light on differences in CI assembly across lineages and suggests potential physiological roles for CI* beyond assembly.

## Introduction

Respiration is an essential metabolic process that provides the energy and intermediate metabolites needed for growth and maintenance of all eukaryotes. In plants, respiratory pathways are not only involved in energy conversion but also play crucial roles in the procurement of biosynthetic precursors and in the balancing of the cellular redox state (*O'Leary et al., 2019*). Plant respiratory processes are also closely intertwined with photosynthetic pathways. Despite the importance of respiratory processes to plants' biomass accumulation, carbon flux and acclimation (*O'Leary et al., 2019*; *Amthor et al., 2019*; *Heskel et al., 2016*), the fundamental mechanisms by which the plant mitochondrial electron transport chain (mETC) produces proton ($H^+$) gradients that are converted into chemical energy remain poorly understood. Molecular knowledge of the structures and mechanisms of the plant mETC components, which differ significantly in their assembly and composition from better-studied mammalian systems, is essential to understand how plants efficiently convert energy and balance respiration with photosynthesis.

Plant mitochondria possess a 'canonical' mETC shared with most eukaryotes that is composed of four large membrane protein complexes (complexes I-IV, CI-IV) and an associated ATP synthase in the inner mitochondrial membrane (IMM). Complexes I-IV couple oxidoreduction reactions to $H^+$ pumping against the concentration gradient across the IMM to produce a large $H^+$ electrochemical potential ('proton motive force') that is then dissipated through ATP synthase's rotary mechanism to produce ATP in the mitochondrial matrix. Additionally, plants also possess an 'alternative' mETC that dissipates reduction equivalents in a non-$H^+$-pumping, non-energy-conserving fashion (*Millar et al., 2011*; *Schertl and Braun, 2014*).

Complex I (CI) is the main energy-conserving entry point for electrons into the mETC. In plants, as in most eukaryotes so far studied, CI is the largest (~1 MDa) and mechanistically least understood

*For correspondence:
jaletts@ucdavis.edu

**Competing interests:** The authors declare that no competing interests exist.

**eLife digest** Respiration is the process used by all forms of life to turn organic matter from food into energy that cells can use to live and grow. The final stage of this process relies on an intricate chain of protein complexes which produce the molecule that cells use for energy. Complexes in the chain are made up of specific proteins that are carefully assembled, often into discrete modules or intermediate complexes, before coming together to form the full protein complex. Understanding how these complexes are assembled provides important insights into how respiration works.

The precise three-dimensional structure of these complexes has been identified for bacteria, yeast and mammals. However, less is known about how these respiration complexes form in plants. For this reason, Maldonado et al. studied the structure of an intermediate complex that is only found in plants, called CI*. This intermediate structure goes on to form complex I – the largest complex in the respiration chain.

A technique called cryo-electron microscopy was used to obtain a structure of CI* at a near-atomic level of detail. This structure revealed how the proteins that make up CI* fit together, highlighting differences and similarities in how plants assemble complex I compared to bacteria, yeast and mammals. Maldonado et al. also studied the activity of CI*, leading to the suggestion that this complex may be more than just a stepping stone towards building the full complex I and could have its own role in the cell.

The structure of this complex provides new insights into the respiration mechanism of plants and could help scientists improve crop production. For instance, new compounds may be able to block respiration in pests, while leaving the crop unharmed; or genetic modifications could create plants that respire more efficiently in different environments.

component of the mETC (*Sazanov, 2015*; *Hirst, 2013*). CI oxidizes NADH and reduces coenzyme Q (CoQ, ubiquinone), pumping four $H^+$ per two electrons from NADH (*Jones et al., 2017*). CI is an L-shaped multiprotein complex, with a membrane arm and a peripheral arm. In eukaryotes, the peripheral arm of CI extends into the mitochondrial matrix, while the membrane arm is buried within the IMM. Both arms are composed of 'modules' with specific functions and distinct evolutionary origins (*Efremov and Sazanov, 2012*). The peripheral arm contains the NADH dehydrogenase N-module and the CoQ-reducing Q-module, which provide the binding sites for NADH and quinone, respectively, as well as the chain of FeS clusters needed for electron transfer (*Figure 1A*). The membrane arm contains four proton pumps, two of which are located in the proximal-pumping module ($P_P$), with the remaining two pumps in the distal-pumping module ($P_D$; *Figure 1A*; *Dröse et al., 2011*). Through a still poorly understood mechanism, the energy released from NADH-CoQ oxidoreduction in the peripheral arm (N- and Q-modules) is coupled to conformational changes along the membrane arm ($P_P$ and $P_D$), resulting in proton pumping from the mitochondrial matrix into the mitochondrial intermembrane space (IMS).

Across the studied eukaryotes, mitochondrial CI is composed of 14 highly conserved 'core' subunits that are responsible for electron transport and $H^+$ pumping, and 30–35 'accessory' subunits that are involved in CI's assembly, stability and regulation (*Millar et al., 2011*; *Meyer, 2012*). The exact number of subunits in plant mitochondrial CI is still unclear, with several mass spectrometry measurements revealing differing compositions (*Meyer, 2012*). Nonetheless, it is known that several plant CI accessory subunits are not found in fungi and metazoans (opisthokonts). Most notably, five gamma-type carbonic anhydrase (γCA) proteins (CA1, CA2, CA3, CAL1, and CAL2) have been shown to be associated with CI in plants (*Sunderhaus et al., 2006*; *Perales et al., 2004*). These proteins are located on the matrix side of CI's membrane arm, likely as a heterotrimer of CAL1 or CAL2 monomer plus a CA1/CA2 hetero- or homodimer (*Fromm et al., 2016*). Hence, only a subset of the five γCA proteins are expected to be simultaneously associated with CI. Although the exact γCA protein combinations are likely tissue- and development-stage-dependent (*Córdoba et al., 2019*), the role of the γCA domain in plant CI's function is unknown (*Martin et al., 2009*).

Another major difference between plants and metazoans occurs in the CI assembly pathway. In metazoans, the N-module (which is responsible for NADH oxidation) is assembled onto the rest of the complex (Q-, $P_P$- and $P_D$-modules) as the final step of assembly (*Formosa et al., 2018*;

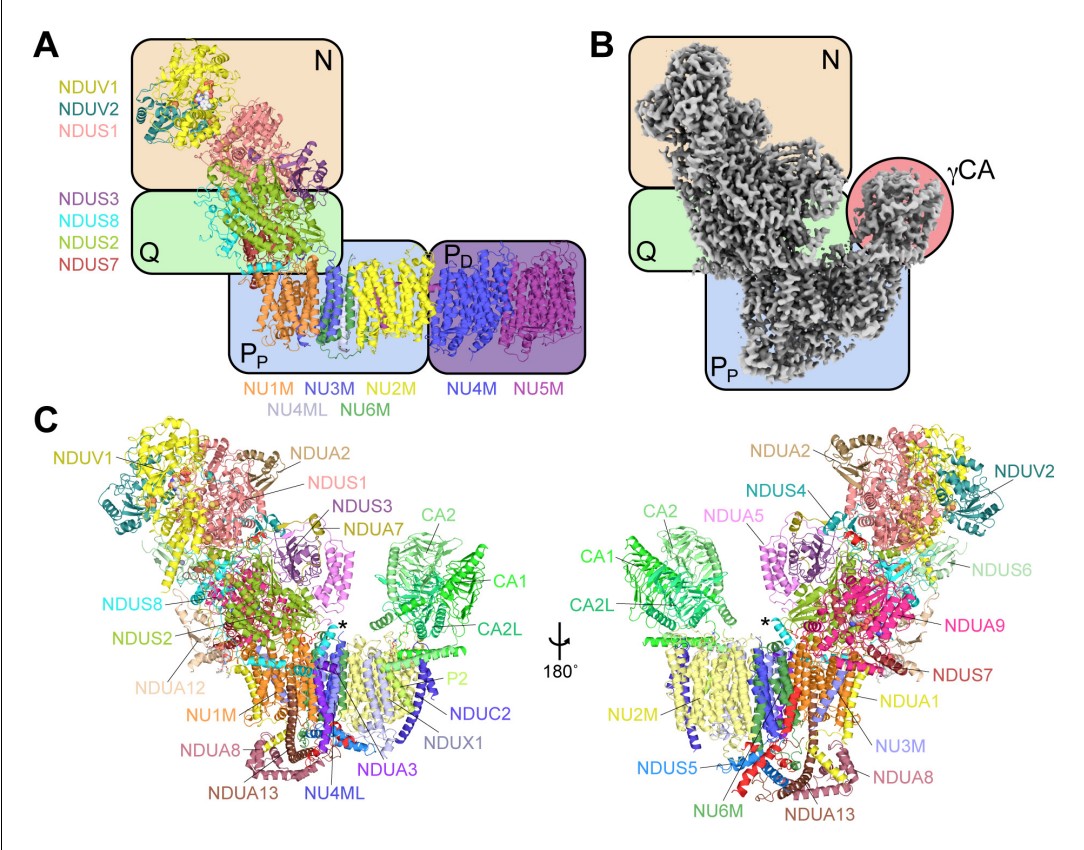

**Figure 1.** The structure of CI* from *Vigna radiata*. (A) An overview of the conserved modular structure of CI using the *Thermus thermophilus* bacterial core subunits as a simple model (PDB: 4HEA) (*Baradaran et al., 2013*). (B) CryoEM density map of CI* from *V. radiata* highlighting its modular architecture. N, NADH-binding module; Q, quinone-binding module; $P_P$, proximal-pump module; $P_D$, distal-pump module; γCA, carbonic anhydrase domain, see also *Video 1*). (C) Atomic model of *V. radiata* CI* with all 30 assigned subunits labeled. The additional N-terminal helix of NDUS8 is indicated with an asterisk (*).

The online version of this article includes the following figure supplement(s) for figure 1:

**Figure supplement 1.** Schematic CI assembly pathways in metazoans and plants.

**Figure supplement 2.** Purification and characterization of CI*.

**Figure supplement 3.** CryoEM processing steps.

**Figure supplement 4.** CryoEM model-to-map correlation.

*Guerrero-Castillo et al., 2017*; *Garcia et al., 2017*; *Stroud et al., 2016*; *Figure 1—figure supplement 1A*). In plants, more similar to what occurs in bacterial CI assembly (*Friedrich et al., 2016*), the final assembly step is the attachment of the $P_D$-module onto an intermediate (termed CI*) that already contains the N-, Q- and $P_P$-modules (*Ligas et al., 2019*; *Figure 1—figure supplement 1B*). This difference in the order of assembly of CI in plants vs. metazoans is significant: in metazoans, adding the NADH dehydrogenase N-module last ensures that no assembly intermediate is capable of transferring electrons from NADH to CoQ. This is believed to have protective roles, to prevent the formation of reactive oxygen species during the CI assembly process (*Parey et al., 2019*). In contrast, the plant CI* intermediate contains all the subunits and co-factors needed to carry out NADH: CoQ oxidoreduction.

In contrast to the large number of recent high-resolution structures of mammalian and yeast respiratory complexes and supercomplexes, the most detailed plant CI structures known were obtained by negative-stain electron microscopy (EM) two-dimensional (2D) classifications from *Solanum tuberosum* (potato) and *Arabidopsis thaliana* or sub-tomogram averaged reconstructions that lack secondary structure details (*Bultema et al., 2009*; *Dudkina et al., 2005*; *Davies et al., 2018*). The paucity of functional and structural data for plant mETC complexes stems in large part from the

limited availability of sufficient protein sample needed for structural analysis (*Dudkina et al., 2015*). Indeed, it has been difficult to obtain intact plant mitochondria in sufficient amounts for preparative biochemical fractionation. A typical reported yield of mitochondria is ~0.2–0.5 mg mitochondria/g fresh weight of starting plant material (*Luster and Fites, 1987*), which contrasts with a yield of ~30 mg mitochondria/g fresh weight from mammalian sources. In light of these challenges, most of the biochemical data on plant mETC have used intact mitochondria (e.g. oxygen-consumption experiments) or complexes that have been electro-eluted from electrophoretic gels (*Bultema et al., 2009*; *Dudkina et al., 2005*; *Dudkina et al., 2006*; *Eubel et al., 2005*). Although such electro-eluted protein samples have yielded the low-resolution structures described above and have proven suitable for proteomic studies, the low yields and low activities of these protein samples have so far thwarted detailed functional or structural analyses of the plant mETC complexes. A detailed understanding of the energy-converting mechanisms of plant respiratory mETC complexes and supercomplexes requires improved protocols for their extraction from plant mitochondrial membranes, and their purification in sufficient amounts while maintaining them in a functionally active state.

Here, we present a cryoEM structure of an ~800 kDa assembly intermediate of plant mitochondrial CI from etiolated *Vigna radiata* (mung bean) hypocotyls at 3.9 Å resolution. This assembly intermediate, CI* (*Ligas et al., 2019*), contains the intact peripheral arm (N- and Q-modules) as well as the $P_P$-module and γCA domain, but lacks the $P_D$-module. Our structure allowed us to build the first atomic model for any mitochondrial CI species from the plant kingdom and revealed important differences in the CI core and accessory subunits between plants, mammals, yeast and bacteria. Such subunit differences shed light on the known differences in CI assembly in plants versus opisthokonts. The structure also allowed us to define the interface between the γCA domain and the membrane arm of CI and revealed a key role for lipids in this interaction. We also discuss the implications of our findings on the possibility that CI* may provide additional flexibility to plants' mETC.

## Results

### Structure of a mitochondrial Complex I assembly intermediate from etiolated *V. radiata* (mung bean)

In order to investigate the plant mitochondrial electron transport chain, we identified *V. radiata* (mung bean) as an optimal model system. *V. radiata* offers several advantages for plant mitochondrial research: i) it can be easily sprouted and harvested within six days, ii) it can be grown in the dark (etiolated) to minimize development of chloroplasts, which would otherwise contaminate the mitochondrial preparations, iii) its age and growth conditions can be controlled experimentally, iv) its genome has been sequenced and v) its mitochondrial content has been reported to be higher than other plant sources previously used for plant mitochondrial research (*Luster and Fites, 1987*). Moreover, we have optimized standard plant mitochondria isolation protocols (*Millar et al., 2007*) to routinely obtain ~1 g of wet weight mitochondria per 1 kg of etiolated *V. radiata* hypocotyls, approximately 3–4 times what has been previously reported (*Luster and Fites, 1987*).

Isolation of the mitochondrial electron transport complexes of *V. radiata* was performed by extraction from washed mitochondrial membranes using the gentle detergent digitonin, followed by exchange into the amphipathic polymer A8-35 to further stabilize the complexes. The presence of complex I (CI)-containing bands was analyzed using a standard in-gel NADH-dehydrogenase activity assay for CI on a blue-native gel (BN-PAGE) (*Schertl and Braun, 2015*). As expected from previously reported plant mitochondrial extractions (*Bultema et al., 2009*; *Dudkina et al., 2005*; *Eubel et al., 2004a*; *Eubel et al., 2004b*; *Eubel et al., 2003*; *Krause et al., 2004*), we observed a number of bands with NADH-dehydrogenase activity, representing CI in different assembly states, such as in mitochondrial supercomplexes (*Bultema et al., 2009*; *Dudkina et al., 2005*; *Eubel et al., 2004a*; *Eubel et al., 2004b*; *Eubel et al., 2003*; *Krause et al., 2004*; *Dudkina et al., 2010*; *Figure 1—figure supplement 2A*). The amphipol-stabilized complexes and supercomplexes were separated on a linear sucrose gradient (*Figure 1—figure supplement 2B–C*). Two peaks displaying NADH-dehydrogenase activity were of sufficient amount to be further purified by size-exclusion chromatography (*Figure 1—figure supplement 2D*). These purified fractions retained their NADH-dehydrogenase activity by in-gel activity assays (*Figure 1—figure supplement 2E*). Moreover, these fractions also showed NADH-decylubiquinone oxidoreductase activity using a standard CI spectroscopic activity

assay (*Huang et al., 2015*; *Figure 1—figure supplement 2F*). These fractions were investigated by single-particle cryoEM. Here, we present results from the lower molecular weight fraction ('peak 2') (*Figure 1—figure supplement 2G–H*).

Structural analysis revealed that this fraction contained an ~800 kDa CI subcomplex, previously identified as a plant mitochondrial CI assembly intermediate termed complex I* (CI*, *Figure 1B*), which we were able to resolve to a nominal resolution of 3.9 Å (*Figure 1C*, *Tables 1–2*, *Video 1*). The existence of this assembly intermediate has been determined by genetic and mitochondrial pro-teomics experiments of CI's assembly pathway in etiolated seedlings (*Heazlewood et al., 2003*) and non-etiolated seedlings and leaves of *Arabidopsis thaliana* (*Ligas et al., 2019*; *Meyer et al., 2011*; *Schertl et al., 2012*; *Schimmeyer et al., 2016*; *Senkler et al., 2017*), as well in non-etiolated leaves of *Nicotiana sylvestris* (*Pineau et al., 2008*). Moreover, the *A. thaliana* and *N. sylvestris* CI* interme-diate shows NADH-dehydrogenase activity by the same in-gel activity assay used in our preparation (*Meyer et al., 2011*; *Pineau et al., 2008*; *Haili et al., 2013*). CI* contains CI's intact peripheral arm (N- and Q-modules), $P_P$-module and γCA domain. However, it is missing the two membrane arm core subunits NU4M and NU5M and their associated accessory subunits that form the $P_D$-module (*Figure 1B*). As expected from complexome profiling analyses (*Ligas et al., 2019*; *Senkler et al., 2017*), our structure of CI* is composed of over 30 subunits of the N-module, Q-module, $P_P$-module and the γCA domain. Throughout this manuscript, we use the plant nomenclature for the subunits (see *Table 3* for subunit name conversions).

## Key differences in observed core subunits

The peripheral and membrane arm core subunits present in the structure of CI* are structurally homologous to the bacterial, yeast and mammalian CI core subunits, with a few notable differences.

The N-terminus of core Q-module subunit NDUS2 is shortened in *V. radiata* compared to NDUS2 from *Y. lipolytica* and mammals, in which the N-terminus of NDUS2 extends from the interface of the peripheral and membrane arms of the complex along the matrix side of the membrane arm. Whereas in *Y. lipolytica* the N-terminus of NDUS2 binds to the matrix surface of core $H^+$-pumping subunit NU2M, in mammals the N-terminus of NDUS2 extends further along the membrane arm and binds to the matrix surface of core $H^+$-pumping subunit NU4M, bridging across the $P_P$- and $P_D$-mod-ules. In contrast, *V. radiata* NDUS2 is ~40 amino acid residues shorter on the N-terminus compared to mammals and does not extend along the membrane arm. Moreover, the equivalent path for the *Y. lipolytica* or mammalian NDUS2 N-terminus in *V. radiata* is blocked by the γCA domain to the plant $P_P$-module on the membrane arm.

The N-terminus of core peripheral arm subunit NDUS8 is also divergent between plants, fungi and mammals. In *V. radiata*, the N-terminus possesses an additional α-helix that binds between the Q-module accessory subunit NDUA5 and the $P_P$-module core membrane subunit NU2M, enlarging the interaction interface between the peripheral and membrane arms (*Figure 1C*). In *Y. lipolytica*, the N-terminus of NDUS8 forms an extended coil that reaches up along the peripheral arm between the Q-module accessory subunits NDUA5 and NDUA7, making contact with the core Q-module sub-unit NDUS3. In contrast, the N-terminus of mammalian NDUS8 folds back along the surface of the membrane arm and tucks underneath the Q-module accessory subunit NDUA7. In *Y. lipolytica*, this binding site, underneath the NDUA7 homologue (NUZM), is occupied by NUZM's C-terminus, which folds back under itself. However, in *V. radiata* the binding site underneath NDUA7 is occupied by an unidentified subunit that extends from this pocket under NDUA7 toward the core transmembrane subunits adjacent to the NU3M transmembrane helix (TMH) 1–2 loop and the NU6M TMH3, which undergo conformational changes during CI's enzymatic turnover in the fungal structures (*Agip et al., 2018*; *Letts et al., 2019*; *Parey et al., 2018*). Although the identity of this sequence in the *V. radiata* structure remains unclear, it appears to be unique to plant CI.

Core subunit NU2M in *V. radiata* CI* contains three N-terminal transmembrane helices that are present in yeast and bacterial complexes, but lost in the metazoan lineage (*Birrell and Hirst, 2010*). Moreover, *V. radiata* CI* contains a homologue of *Y. lipolytica*'s accessory subunit NUXM (absent in metazoans), which binds to the NU2M N-terminal transmembrane helices. Based on the *Y. lipolytica* subunit name, we coined this subunit of *V. radiata* CI NDUX1. The presence of this subunit in both plants and fungi suggests that this subunit was present in the ancestral eukaryotic CI before the uni-kont/bikont lineage divergence but was lost in metazoans when NU2M became N-terminally trun-cated. The first transmembrane helix of NU2M in *Y. lipolytica* is notably short (only 15 amino acids),

**Table 1.** Cryo-EM data collection, refinement and validation statistics.

**Data Collection and processing**

| Microscope | Titan krios, (UCSF) |
|---|---|
| Camera | K3 |
| Magnification | 60,010 |
| Voltage (kV) | 300 kV |
| Electron exposure (e⁻/Å²) | 86.4 |
| Defocus range (μm) | −0.5 to −2.0 |
| Pixel size (Å) | 0.8332 |
| Software | SerialEM |

| Reconstruction | CI* Peripheral Arm | CI* Membrane Arm | CI* Composite Map |
|---|---|---|---|
| Number of particles | 34,407 | 34,407 | |
| Accuracy of rotations (°) | 0.68 | 1.489 | |
| Accuracy of translations (pixels) | 0.655 | 0.881 | The CI* Peripheral Arm and Membrane Arm Maps were combined in Phenix to generate this composite map |
| Box size (pixels) | 512 | 512 | |
| Final resolution (Å) | 3.8 | 3.9 | |
| Map sharpening B factor (Å²) | −90 | −96 | |
| EMDB ID | 22093 | 22092 | 22090 |

**Refinement**

| Software | Phenix | | |
|---|---|---|---|
| Initial model (PDB code) | 6Q9D | 6Q9B and 1QRG | 6Q9D, 6Q9B and 1QRG |
| **Map/model correlation** | | | |
| Model resolution (Å) | 3.9 | 4.0 | 3.9 |
| d99 (Å) | 3.9 | 4.0 | 4.0 |
| FSC model 0.5 (Å) | 3.9 | 3.9 | 3.9 |
| Map CC (around atoms) | 0.82 | 0.86 | 0.87 |
| **Model composition** | | | |
| Non-hydrogen atoms | 26,001 | 19,052 | 45047 |
| Protein residues | 3284 | 2453 | 5736 |
| Number of chains | 17 | 18 | 34 |
| Number of ligands and cofactors | 11 | 1 | 12 |
| Number of lipids | 0 | 6 | 6 |
| **Atomic Displacement Parameters (ADP)** | | | |
| Protein average (Å²) | 68.78 | 58.40 | 64.39 |
| Ligand average (Å²) | 48.59 | 48.59 | 48.59 |
| **R.m.s. deviations** | | | |
| Bond lengths (Å) | 0.007 | 0.007 | 0.007 |
| Bond angles (°) | 1.187 | 1.122 | 0.845 |
| **Ramachandran Plot** | | | |
| Favored (%) | 82.90 | 88.03 | 84.98 |
| Allowed (%) | 16.76 | 11.88 | 14.79 |
| Disallowed (%) | 0.34 | 0.08 | 0.23 |
| **Validation** | | | |
| MolProbity score | 2.41 | 2.31 | 2.38 |

*Table 1 continued on next page*

| | | | |
|---|---|---|---|
| Clash score | 16.79 | 16.21 | 16.42 |
| Rotamer outliers (%) | 0.25 | 0.20 | 0.23 |
| EMRinger score | 1.47 | 2.09 | 2.17 |
| PDB ID | — | — | 6X89 |

enters only to the midplane of the membrane and is bound by a membrane-penetrating loop of the accessory subunit NUXM. In contrast, in bacteria (*T. thermophilus* and *E. coli*) and *V. radiata*, the first transmembrane helix of NU2M spans the full length of the membrane. Furthermore, the loop connecting *V. radiata*'s NU2M TMH1-2 in the mitochondrial matrix is longer than in any of the other CI structures and extends into the matrix, where it contacts the N-terminal helix of NDUS8 discussed above. Given the universality of the hinging motion between CI's peripheral and membrane arms, seen in the structures of several organisms (*Agip et al., 2018*; *Letts et al., 2019*; *Parey et al., 2018*), the additional interaction surface formed by NDUS8 and NU2M in *V. radiata* CI is likely functionally relevant.

## Key differences in observed accessory subunits

Although the majority of the accessory subunits present in CI* have homologues in fungi and mammals (opisthokonts), there are a number of notable differences.

In the plant complex, the peripheral arm accessory subunit NDUS6 lacks an N-terminal domain that is seen in both the *Y. lipolytica* and mammalian structures (*Figure 2A*). In *Y. lipolytica*, mammals and *V. radiata*, the C-terminal, $Zn^{2+}$-containing domain of NDUS6 binds mainly to the core subunits NDUS1, NDUS8 and NDUS2 at the interface of the N- and Q-modules. However, in opisthokonts, the N-terminal domain of NDUS6 binds to the Q-module at an additional site through contacts with the membrane-anchored NDUA9 accessory subunit (*Figure 2A*). In order to bind across these two locations, NDUS6 in opisthokonts extends above the C-terminus of accessory subunit NDUA12. This arrangement determines the order of assembly of these subunits in opisthokonts, as NDUA12 must be bound to the peripheral arm before the N-terminal domain of NDUS6 binds. However, due to the lack of the N-terminal domain in *V. radiata*'s NDUS6, there is no interaction with NDUA9 nor traversing of the NDUA12 C-terminus. This difference has important implications for the assembly of CI in plants versus opisthokonts. In opisthokonts, the interaction between NDUS6, NDUA12 and the NDUA12-homologous assembly factor NDUFAF2 establishes an important checkpoint for assembly of the peripheral arm. Thus, the lack of the NDUS6 N-terminus may in part explain observed differences between the assembly pathways of plant and opisthokont CI (see Discussion).

Other key differences can be seen on the intermembrane space side of the membrane arm in accessory subunits NDUA8 and NDUC2. Compared to both *Y. lipolytica* and mammals, the double-CHCH domain of the $P_P$-module NDUA8 subunit, which binds to the 'heel' of the complex on the intermembrane space (*Figure 2B*), is C-terminally truncated in *V. radiata*. In the *Y. lipolytica* structure, the C-terminus of NDUA8 folds back onto itself with an additional α-helix, forming a bulkier subunit and a further interaction interface with the core transmembrane subunit NU1M. More interestingly, in mammals, the C-terminus of NDUA8 extends as a long coil halfway along the membrane arm and binds in a pocket between NU2M and NU4M at the interface of the $P_P$-module and $P_D$-module. The $P_P$-module accessory subunit NDUC2 is also C-terminally truncated in *V. radiata* and *Y. lipolytica* relative to NDUC2 in mammals (*Figure 2C*). In all mitochondrial CI structures to date, this subunit binds to the final transmembrane helix of the core NU2M subunit. However, in mammals, the NDUC2 C-terminus forms an extended coil on the intermembrane space side of the complex that extends along the membrane arm to interact with NDUB10 and NDUB11, bridging the $P_P$- and $P_D$- modules. This bridging interaction is also present in *Y. lipolytica* via an extended loop on the $P_D$-module core subunit NU4M.

This pattern of truncated core and accessory subunits or missing interactions (e.g. NDUS2, NDUA8 and NDUC2; *Table 4*) in *V. radiata* relative to those in opisthokonts likely diminishes the stability of the attachment of $P_P$-module to the $P_D$-module, which may have consequences for CI's function and assembly (see Discussion).

**Table 2.** Model building statistics by subunit.

| Subunit name | Uniprot ID | Chain ID | Total residues | Atomic residues | Poly-Ala | Un-modeled residues | % atomic | TMH | Identified RNA editing sites* | Ligands, lipids |
|---|---|---|---|---|---|---|---|---|---|---|
| **Peripheral arm core subunits** | | | | | | | | | | |
| NDUS1 | A0A1S3TQ85 | S1 | 746 | 57–744 | 57 | 1–56, 745–746 | 92.1% | | | 4Fe4S×2, 2Fe2S |
| NDUS2 | E9KZN6 | S2 | 394 | 9–17,21-394 | | 1–8, 18–20 | 98.0% | | S26L, 246L, S67F, H82Y, S84L, R106C, S112L, S193L, S233L, H242Y, S245L, P247F, R257C, R353C, S360F, S363L, S368F, P375L | |
| NDUS3 | E9KZM7 | S3 | 190 | 1–184 | | 185–190 | 96.8% | | S31F, S56L, P100S, R110W, S133L, L147F | |
| NDS7 | A0A1S3U8J5 | S7 | 213 | 56–213 | | 1–55 | 74.2% | | | 4Fe4S, PC |
| NDS8 | A0A1S3VGS8 | S8 | 222 | 42–222 | | 1–41 | 81.5% | | | 4Fe4S×2 |
| NDUV1 | A0A1S3V7V2 | V1 | 491 | 59–491 | | 1–58 | 88.2% | | | 4Fe4S, FMN |
| NDUV2 | A0A1S3U769 | V2 | 251 | 28–243 | | 1–27, 244–251 | 86.1% | | | 2Fe2S |
| **Peripheral arm accessory subunits** | | | | | | | | | | |
| NDUA2 | A0A1S3TVC7 | A2 | 98 | 2–93 | | 1, 94–98 | 93.9% | | | |
| NDUA5 | A0A1S3U023 | A5 | 169 | 12–137 | | 1–11, 138–169 | 74.6% | | | |
| NDUA6 | A0A1S3W1K8 | A6 | 132 | 118–131 | | 1–117, 132 | 11.4% | | | |
| NDUA7 | A0A1S3UVC7 | A7 | 127 | 19–127 | | 1–18 | 85.8% | | | |
| NDUA9 | A0A1S3V8W7 | A9 | 396 | 47–381 | | 1–46, 382–396 | 84.6% | | | NADPH |
| NDUA12 | A0A1S3VNK7 | AL | 156 | 21–155 | | 1–20, 156 | 86.5% | | | |
| NDUS4 | A0A1S3UIW7 | S4 | 146 | 42–142 | 142 | 1–41, 143–146 | 69.2% | | | |
| NDUS6 | A0A1S3VYF3 | S6 | 103 | 31–102 | | 1–30, 103 | 69.9% | | | Zn$^{2+}$ |
| **Membrane arm core subunits** | | | | | | | | | | |
| NU1M | A0A1S4ETV6/ E9KZL0 | 1M | 325 | 2–213, 220–325 | | 1, 214–219 | 97.8% | 8 | R89W, P164S, R165C, S167L, S179F, R225C, P242L, P248L, P252L, R300W, R310W | |
| NU2M | E9KZK9 | 2M | 488 | 1–487 | | 488 | 99.8% | 14 | S19F, S103F, S104F, P119L, P121S, R123C, H132Y, P143L, S166LL, S221F, P307L, H310Y, R320C, S376L, S467L, S468F, S486L | PC×2 |
| NU3M | Q9XPB4 | 3M | 118 | 1–28, 56–118 | | 29–55 | 77.1% | 3 | P70F, P83S, P84L, S115L, R117W | |
| NU4LM | A0A1S4ETY3/ E9KZN8 | 4L | 100 | 1–86 | | 87–100 | 86.0% | 3 | S14F, P29L, S32L, P34S, S37L, S53L, S63L, S66L | |
| NU6M | E9KZM5 | 6M | 205 | 1–72, 111–172 | 73–110 | 173–205 | 65.4% | 5 | P9L, A18V, P30F, P32L, R35C, P54L, H57Y | |
| **Membrane arm accessory subunits** | | | | | | | | | | |
| CA1 | A0A1S3VT00 | G1 | 270 | 3–222 | 223–233 | 1–2, 234–270 | 81.5% | | | |
| CA2 | A0A1S3U544 | G2 | 273 | 2–237 | | 1, 238–273 | 86.4% | | | |
| CAL2 | A0A1S3UI49 | L2 | 256 | 49–129, 134–254 | | 1–48, 130–133, 255–256 | 80.5% | | | |
| NDUX1 | A0A1S3VI15 | X1 | 101 | 1–100 | | 101 | 99.0% | 2 | | |
| NDUC2 | A0A1S3UPL8 | C2 | 81 | 5–68 | | 1–4, 69–81 | 79.0% | 2 | | |
| NDUA8 | A0A1S3VVN6 | A8 | 106 | 2–106 | | 1 | 99.1% | | | |
| NDUA13 | A0A1S3UYW0 | AM | 143 | 2–143 | | 1 | 99.3% | 1 | | |

*Table 2 continued on next page*

*Table 2 continued*

| Subunit name | Uniprot ID | Chain ID | Total residues | Atomic residues | Poly-Ala | Un-modeled residues | % atomic | TMH | Identified RNA editing sites* | Ligands, lipids |
|---|---|---|---|---|---|---|---|---|---|---|
| NDUA1 | A0A1S3TU57 | A1 | 65 | 2–63 | | 1, 64–65 | 95.4% | 1 | | PC |
| NDS5 | A0A1S3TQ33 | S5 | 399 | 2–70 | | 1, 71–399 | 17.3% | | | |
| NDUA3 | A0A1S3TCK0 | A3 | 63 | 2–45 | | 1, 46–63 | 69.8% | 1 | | |
| P2 | A0A1S3TGE7 | P2 | 115 | 83–106 | 77–82 | 1–76, 107–115 | 20.9% | | | |
| **Unassigned density** | | | | | | | | | | |
| | | A | | | 1–18 | | | | | |
| | | B | | | 1–24 | | | | | |
| | | C | | | 1–43 | | | 1 | | |

*RNA editing of mitochondrially encoded subunits: amino acids were changed at the listed positions as detailed. The changes were based on the reported equivalent *A. thaliana* RNA edits (**Giegé and Brennicke, 1999**; **Bentolila et al., 2008**) and were only made when density was unambiguously correct for the edited *V. radiata* amino acid in the cryoEM map.

## Known Q-module accessory subunits not present in CI*

Compared to the mammalian and *Y. lipolytica* structures, two accessory subunits are absent from the Q-module in the CI* structure, namely the LYR-protein subunit NDUA6 and its accompanying acyl-carrier protein (ACPM1). The absence of the NDUA6 and ACPM1 subunits in CI* is notable given that, when the *Y. lipolytica* NDUA6 homologue is knocked out or mutated, this severely impacts the activity of the complex (**Angerer et al., 2014**). Therefore, although it is not completely understood how NDUA6 modulates the activity of CI, the lack of NDUA6 in CI* may be a way to regulate the activity of the assembly intermediate.

Although densities for NDUA6 and ACPM1 are absent in our CI* structure, density can be seen for a short α-helix bound under NDUS1, where the C-terminus of NDUA6 binds in both the *Y. lipolytica* and mammalian structures. This suggests that NDUA6 may be bound to CI* via its C-terminus, without fully engaging with the complex. Although this would be surprising, the density for the amino acid sidechains in this region is consistent with the sequence of the NDUA6 C-terminus; thus, this density was modelled as such. If correct, this suggests that NDUA6 may be attached to the Q module but unable to fully bind to its main site on NDUS2.

## Plant-specific accessory subunits

*V. radiata* CI* does not have any plant-specific accessory subunits on the peripheral arm. Notwithstanding the unique features of NDUS6 and the absence of NDUA6 and ACPM1 discussed above, all of the *V. radiata* CI* N- and Q-module subunits have homologues in fungi and metazoans. However, this is not the case for the $P_P$-module. Most notably, a large (~90 kDa) heterotrimeric γCA domain lies on top of the core membrane arm subunit NU2M (*Figure 1C*).

The identity of the components of the plant γCA has remained elusive, with different three-way combinations of the five plant γCA proteins proposed based on different genetic and biochemical studies (**Sunderhaus et al., 2006**; **Perales et al., 2004**; **Fromm et al., 2016**; **Cïˉ Rdoba et al., 2019**). Our structure allowed us to unambiguously assign the identity of the subunits of the γCA domain despite high sequence identity between the five carbonic anhydrase

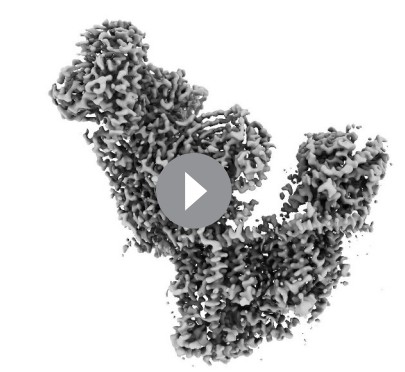

**Video 1.** CryoEM density for the CI* composite map.
https://elifesciences.org/articles/56664#video1

**Table 3.** Complex I subunit homologues in plants, mammals, yeast and bacteria.

*V. radiata* homologues were obtained by performing BLASTp searches of the *Arabidopsis thaliana* genes (**Meyer et al., 2019**; **Braun et al., 2014**). Mammalian, yeast and bacterial homologues were obtained from **Letts and Sazanov, 2015**. Additional BLASTp searches were performed wherever necessary. Given the high sequence similarity between the carbonic anhydrase (CA) paralogues, the names of the *V. radiata* CA proteins appear to have been mis-assigned in the genetic databases relative to their *A. thaliana* homologues. The CA1, CA2, CA2-like nomenclature used in the table is the one that, based on our sequence alignments, best represents homology to the *A. thaliana* CA proteins. N, NADH-binding module; Q, quinone-binding module; $P_P$, proximal-pumps module; $P_D$, distal-pumps module; CA, carbonic anhydrase domain.

| Module | Vigna radiata protein name | Vigna radiata gene | Vigna radiata uniprot identifier | Arabidopsis thaliana protein name | Arabidopsis thaliana gene | Homo sapiens name | Ovis aries name | Mus musculus name | Yarrowia lipolytica name | Thermus thermophilus name |
|---|---|---|---|---|---|---|---|---|---|---|
| **CORE peripheral arm** | | | | | | | | | | |
| N | NDUS1 | LOC106757688 | A0A1S3TQ85 | 75 kDa | At5g37510 | NDUFS1 | NDUFS1 | NDUFS1 | NUAM | Nqo3 |
| N | NDUV1 | LOC106772405 | A0A1S3V7V2 | 51 kDa | At5g08530 | NDUFV1 | NDUFV1 | NDUFV1 | NUBM | Nqo1 |
| N | NDUV2 | LOC106762461 | A0A1S3U769 | 24 kDa | At4g02580 | NDUFV2 | NDUFV2 | NDUFV2 | NUHM | Nqo2 |
| Q | NDUS2 | nad7 | E9KZN6 | Nad7 | AtMg00510 | NDUFS2 | NDUFS2 | NDUFS2 | NUCM | Nqo4 |
| Q | NDUS3 | nad9 | E9KZM7 | Nad9 | AtMg00070 | NDUFS3 | NDUFS3 | NDUFS3 | NUGM | Nqo5 |
| Q | NDS7 | LOC106762764 | A0A1S3U8J5 | PSST | At5g11770 | NDUFS7 | NDUFS7 | NDUFS7 | NUKM | Nqo6 |
| Q | NDS8 | LOC106775047 | A0A1S3VGS8 | TYKY | At1g79010, At1g16700 | NDUFS8 | NDUFS8 | NDUFS8 | NUIM | Nqo9 |
| **CORE membrane arm** | | | | | | | | | | |
| $P_P$ | NU1M | nad1 | A0A1S4ETV6 | Nad1 | AtMg00516, AtMg01120, AtMg01275 | MT-ND1 | MT-ND1 | MT-ND1 | NU1M | Nqo8 |
| $P_P$ | NU2M | nad2 | E9KZK9 | Nad2 | AtMg00285, AtMg01320 | MT-ND2 | MT-ND2 | MT-ND2 | NU2M | Nqo14 |
| $P_P$ | NU3M | nad3 | Q9XPB4 | Nad3 | AtMg00990 | MT-ND3 | MT-ND3 | MT-ND3 | NU3M | Nqo7 |
| $P_P$ | NU4LM | nad4L | A0A1S4ETY3 | Nad4L | AtMg00650 | MT-ND4L | MT-ND4L | MT-ND4L | NULM | Nqo11 |
| $P_P$ | NU6M | nad6 | E9KZM5 | Nad6 | AtMg00270 | MT-ND6 | MT-ND6 | MT-ND6 | NU6M | Nqo10 |
| $P_D$ | NU4M* | nad4 | E9KZL8 | Nad4 | AtMg00580 | MT-ND4 | MT-ND4 | MT-ND4 | NU4M | Nqo13 |
| $P_D$ | NU5M* | nad5 | E9KZL1 | Nad5 | AtMg00060, AtMg00513, AtMg00665 | MT-ND5 | MT-ND5 | MT-ND5 | NU5M | Nqo12 |
| **ACCESSORY membrane arm** | | | | | | | | | | |
| $P_P$ | NDUA1 | LOC106758834 | A0A1S3TU57 | MWFE | At3g08610 | NDUFA1 | NDUFA1 | NDUFA1 | NIMM | - |
| $P_P$ | NDUA3 | LOC106754061 | A0A1S3TCK0 | B9 | At2g46540 | NDUFA3 | NDUFA3 | NDUFA3 | NI9M | - |
| $P_P$ | NDUA8-B | LOC106778955 | A0A1S3VVN6 | PGIV | At3g06310, At5g18800 | NDUFA8 | NDUFA8 | NDUFA8 | NUPM | - |
| $P_P$ | NDUA13-A | LOC106769964 | A0A1S3UYW0 | B16.6 | At2g33220, At1g04630 | NDUFA13 | NDUFA13 | NDUFA13 | NB6M | - |
| $P_P$ | NDUC1 | LOC106771273 | A0A1S3V2Z3 | - | - | NDUFC1 | NDUFC1 | NDUFC1 | - | - |
| $P_P$ | NDUS5 | LOC106757655 | A0A1S3TQ33 | 15 kDa | At3g62790, At2g47690 | NDUFS5 | NDUFS5 | NDUFS5 | NIPM | - |
| $P_P$ | NDUB8 | LOC106765859 | A0A1S3UJ95 | ASHI | At5g47570 | NDUFB8 | NDUFB8 | NDUFB8 | NIAM | - |
| $P_P$ | NDUB10-B | LOC106774903 | A0A1S3VGT1 | PDSW | At1g49140, At3g18410 | NDUFB10 | NDUFB10 | NDUFB10 | NIDM | - |
| $P_P$ | NDUA11* | LOC106756741 | A0A1S3TLY8 | B14.7 | At2g42210 | NDUFA11 | NDUFA11 | NDUFA11 | NUJM | - |
| Module | Vigna radiata protein name | Vigna radiata gene | Vigna radiata Uniprot identifier | Arabidopsis thaliana protein name | Arabidopsis thaliana gene | Homo sapiens name | Ovis aries name | Mus musculus name | Yarrowia lipolytica name | Thermus thermophilus name |
| **ACCESSORY membrane arm** | | | | | | | | | | |

*Table 3 continued on next page*

| | | | | | | | | | | |
|---|---|---|---|---|---|---|---|---|---|---|
| $P_D$ | NDUB1* | LOC106775330 | A0A1S3VI15 | MNLL | At4g16450 | NDUFB1 | NDUFB1 | NDUFB1 | - | - |
| $P_D$ | NDUC2* | LOC106767534 | A0A1S3UPL8 | B14.5b | At4g20150 | NDUFC2 | NDUFC2 | NDUFC2 | NEBM | - |
| $P_D$ | NDUB2* | LOC106754955 | A0A1S3TFG6 | AGGG | At1g76200 | NDUFB2 | NDUFB2 | NDUFB2 | - | - |
| $P_D$ | NDUB3* | LOC106769121 | A0A1S3UVV0 | B12 | At2g02510, At1g14450 | NDUFB3 | NDUFB3 | NDUFB3 | NB2M | - |
| $P_D$ | NDUB4* | LOC106766640 | A0A1S3ULL3 | B15 | At2g31490 | NDUFB4 | NDUFB4 | NDUFB4 | NB5M | - |
| $P_D$ | NDUB5* | LOC106767179 | A0A1S3UND4 | - | - | NDUFB5 | NDUFB5 | NDUFB5 | NUNM | - |
| $P_D$ | NDUB7* | LOC106770979 | A0A1S3V2B8 | B18 | At2g02050 | NDUFB7 | NDUFB7 | NDUFB7 | NB8M | - |
| $P_D$ | NDUB9* | LOC106760947 | A0A1S3U1J6 | B22 | At4g34700 | NDUFB9 | NDUFB9 | NDUFB9 | NI2M | - |
| $P_D$ | NDUB11* | LOC106771273 | A0A1S3V2Z3 | ESSS | At2g42310, At3g57785 | NDUFB11 | NDUFB11 | NDUFB11 | NESM | - |
| **ACCESSORY peripheral arm** | | | | | | | | | | |
| N | NDUA2 | LOC106759195 | A0A1S3TVC7 | B8 | At5g47890 | NDUFA2 | NDUFA2 | NDUFA2 | NI8M | - |
| N | NDUA12 | LOC106776991 | A0A1S3VNK7 | B17.2 | At3g03100 | NDUFA12 | NDUFA12 | NDUFA12 | N7BM | - |
| N | NDUS4 | LOC106765762 | A0A1S3UIW7 | 18 kDa | At5g67590 | NDUFS4 | NDUFS4 | NDUFS4 | NUYM | - |
| N | NDUS6 | LOC106779709 | A0A1S3VYF3 | 13 kDa | At3g03070 | NDUFS6 | NDUFS6 | NDUFS6 | NUMM | - |
| Q | NDUA5 | LOC106760411 | A0A1S3U023 | B13 | At5g52840 | NDUFA5 | NDUFA5 | NDUFA5 | NUFM | - |
| Q | NDUA6† | LOC106780789 | A0A1S3W1K8 | B14 | At3g12260 | NDUFA6 | NDUFA6 | NDUFA6 | NB4N | - |
| Q | NDUA7 | LOC106768957 | A0A1S3UVC7 | B14.5a | At5g08060 | NDUFA7 | NDUFA7 | NDUFA7 | NUZM | - |
| Q | NDUA9 | LOC106772694 | A0A1S3V8W7 | 39 kDa | At2g20360 | NDUFA9 | NDUFA9 | NDUFA9 | NUEM | - |
| **Plant-specific accessory** | | | | | | | | | | |
| CA | CA1‡ | LOC106778103 | A0A1S3VT00 | Gamma-CA 1 | At1g19580 | - | - | - | - | - |
| CA | CA2§ | LOC106761992, LOC106761993 | A0A1S3U566, A0A1S3U544 | Gamma-CA 2 | At1g47260 | - | - | - | - | - |
| CA | CA2-L¶ | LOC106765552 | A0A1S3UI49 | Gamma CA-like 2 | At3g48680 | - | - | - | - | - |
| CA | CA3* | n.a.** | n.a.** | Gamma-CA 3 | At5g66510 | - | - | - | - | - |
| CA | CA1-L* | n.a.** | n.a.** | Gamma-CA-like 1 | At5g63510 | - | - | - | - | - |
| $P_P$ | NDUX1†† | LOC106775330 | A0A1S3VI15 | 20.9 kDa | At4g16450 | - | - | - | NUXM | - |
| $P_P$ | P2/16 kDA | LOC106755236 | A0A1S3TGE7 | P2 | At2g27730 | - | - | - | - | - |
| **Plant-specific accessory** | | | | | | | | | | |
| Unconfirmed plant CI subunits (not seen in CI*) | MICOS (DUF543) | LOC106779628 | A0A1S3VY06 | MICOS subunit Mic10 | At1g72165 | - | - | - | - | - |
| | Uncharacterized protein LOC106758628 | LOC106758628 | A0A1S3TTD7 | NDU10 | At4g00585 | - | - | - | - | |
| | P1/11 kDA | LOC106761134 | A0A1S3U2B9 | P1 | At1g67350 | - | - | - | - | - |
| | P3 | LOC106755586 | A0A1S3THM0 | P3 | At5g14105 | - | - | - | - | - |
| | P4 | LOC106767179 | A0A1S3UND4 | P4 | At1g67785 | - | - | - | - | - |
| | TIM22−4 × 1 | LOC106779665 | A0A3Q0EN44 | TIM22-4 | At1g18320 | - | - | - | - | - |
| | TIM22−4 × 2 | LOC106779665 | A0A1S3VZ08 | TIM22-1 | At3g10110 | - | - | - | - | - |
| | TIM23-2 | LOC106761237 | A0A1S3U2K1 | TIM23-2 | At1g72750 | - | - | - | - | - |
| | Uncharacterized protein LOC106768488 isoform X4 | LOC106768488 | A0A1S3UST2 | SH3/FCH domain protein | At1g68680 | - | - | - | - | - |

*Table 3 continued on next page*

| UDP-galactose transporter 1 | LOC106762681 | A0A1S3U838 | TPT domain-containing protein | At1g72180 | - | - | - | - | - |
| Gravitropic in the light 1 | LOC106779790 | A0A1S3VYR1 | DUF641 domain-containing protein | At2g28430 | - | - | - | - | - |

*Not seen in CI*.

†Only the C-terminus seen in CI* (see Main body and Discussion).

‡Called gamma carbonic anhydrase one in Uniprot.

§Called gamma carbonic anhydrase 1, mitochondrial in Uniprot (mis-assigned in the database).

¶Called gamma carbonic anhydrase-like 2, mitochondrial in Uniprot (mis-assigned in the database).

**Homologue not found using BLASTp.

††New identified subunit.

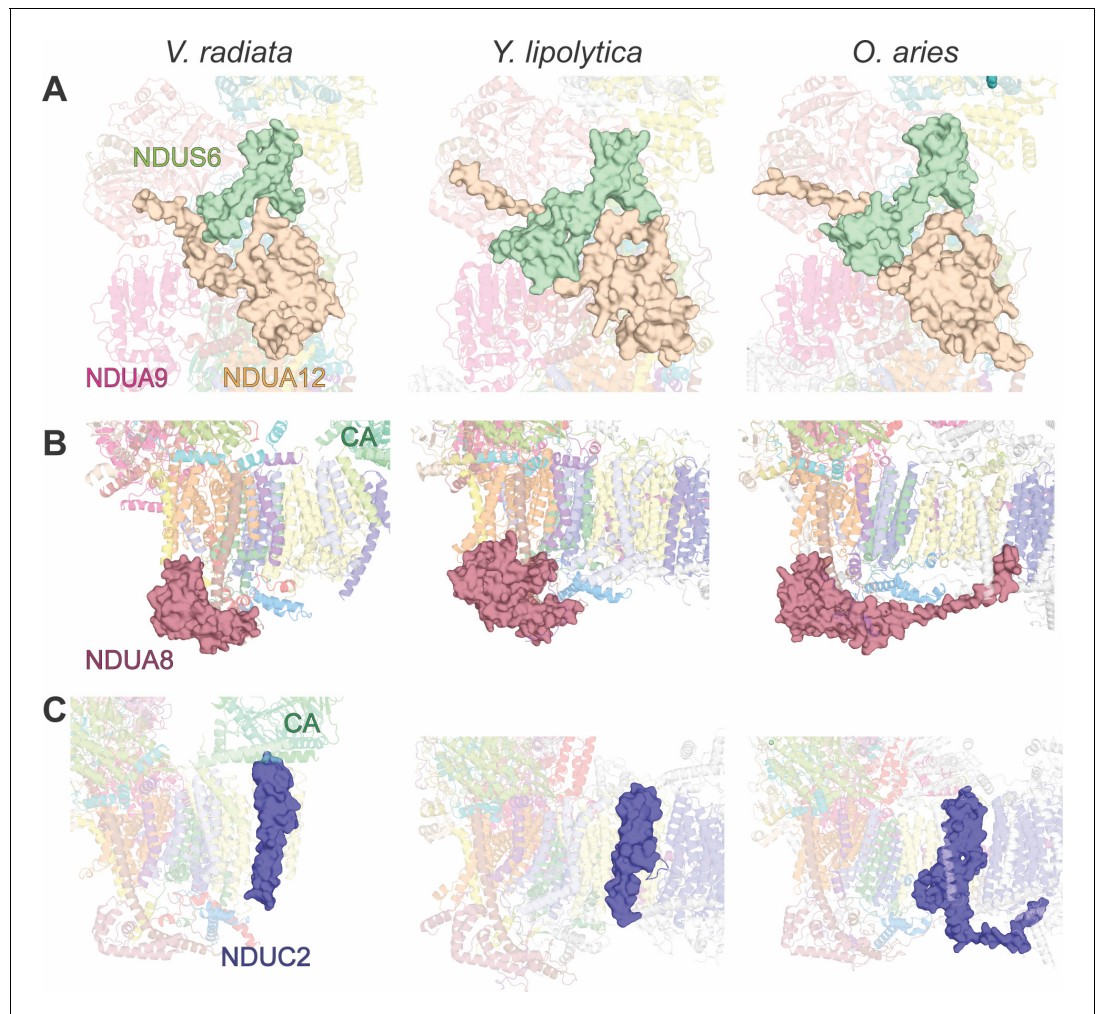

**Figure 2.** Key differences in CI accessory subunits between *V. radiata* and opisthokonts. Accessory subunits NDUS6 and NDUA12, NDUA8 and NDUC2 of *V. radiata* (this study), *Y. lipolytica* (PDB:6RFR) and *O. aries* (PDB: 6QA9) are shown as surface for comparison. (**A**) NDUS6 (green) and NDUA12 (orange), with an additional label for NDUA9. (**B**) NDUA8 (maroon), with additional label for the *V. radiata*'s carbonic anhydrase domain (CA). (**C**) NDUC2 (blue), with additional label for the *V. radiata*'s CA.

**Table 4.** Pₚ- and $P_D$-module bridging subunits in mammalian, *Y. lipolytica* and *V. radiata* CI.
Subunits discussed in the manuscript are marked with two asterisks (**). Bridging interactions are shaded in green. Lack of interactions by existing subunits or lack of homologues are shaded in orange. Lack of the $P_D$ subunits in *V. radiata* CI* is shaded in yellow. Pₚ, proximal pumping domain; $P_D$, distal pumping domain.

| Location | Subunit | Mammals | *Y. lipolytica* | *V. radiata* |
|---|---|---|---|---|
| Inter-membrane space (IMS) | NDUA8** | Extends along membrane arm, bridges NU2M (Pₚ) and NU4M ($P_D$) | Does not extend to the Pₚ/$P_D$-module interface but has an additional helix interacts with NU1M | C-terminally truncated (does not bridge) |
| | NDUC2** | C-terminus bridges NDUB10 (Pₚ) and NDUB11 ($P_D$) | C-terminally truncated, but bridging interaction replaced by extended loop on NU4M | C-terminally truncated (does not bridge) |
| | NDUB5 | Bridging interactions | Bridging interactions | N- and C-terminally truncated (subunit not present in CI*) |
| | NDUA11 | Does not bridge in the IMS | C-terminal extension binding to NU4M | Subunit not present in CI* |
| Membrane | NDUA11 | Binds to the lateral helix of NU5M, connecting NU5M and NU2M | Binds to the lateral helix of NU5M, connecting NU5M and NU2M | Subunit not present in CI* |
| Matrix | NDUS2** | Bridging interactions | Does not bridge | N-terminally truncated (does not bridge) |
| | NU5M | Lateral helix extends into Pₚ | Lateral helix extends into Pₚ | Subunit not present in CI* |
| | NDUA10 | Bridging interactions | No homologue present | No homologue present |
| | NDUB11 | Bridging interactions | Does not bridge | Subunit not present in CI* |
| | NDUB4 | Does not bridge | N-terminus extends along matrix arm and binds to NU2M | N-terminally truncated (subunit not present in CI*) |

proteins in plants. Based on unambiguous density for key non-conserved residues, we were able to definitively assign the three different subunits of *V. radiata* CI* as CA1, CA2 and CA2L (*Figure 3A*).

The interaction surface between the γCA domain and the Pₚ-module (subunits NU2M, NDUC2, P2 and NDUX1) is large, covering an approximate surface of 3,740 Å². As expected (*Sunderhaus et al., 2006*), the γCA interacts with the Pₚ-module tightly, with an approximate gain of solvation free energy of −210 kcal/mol, which is almost twice as large as the solvation energy gain of association of the γCA hetero-trimer itself (*Figure 3A*, *Table 5*).

As has been previously demonstrated by proteomic analysis, the N-terminal mitochondrial signal pre-sequences for CA1 and CA2 remain uncleaved (*Klodmann et al., 2010*). We show here that these two N-terminal sequences together form a short α-helical coiled-coil-like structure (*Figure 3C*). This coiled coil is amphipathic and binds on the matrix surface of the inner mitochondrial membrane, contacting the NDUC2 and P2 subunits (see below) adjacent to the NU2M core subunit. In contrast, no density was observed for the N-terminal pre-sequence of CA2L, consistent with it being post-translationally cleaved (*Huang et al., 2009*).

The physiological role of the γCA domain on plant CI is unknown. Although recombinant mitochondrial γCA from plants has been shown to bind bicarbonate ($HCO_3^-$), it remains unclear whether it exhibits enzymatic activity (*Martin et al., 2009*). The canonical γCA trimer possesses three active sites, one at each interface between two protomers. Each active site is formed by three essential $Zn^{2+}$-coordinating histidine residues. At each active site, two histidine residues are provided by one subunit and the third is provided by the adjacent subunit. However, in the plant CI γCA hetero-trimer, the CA2L subunit is lacking two of the three essential histidine residues (Ala-147 and Arg-152 in *V. radiata*) that would be necessary to form active sites at the interfaces with the CA1 and CA2 subunits. This renders two of the possible three catalytic sites non-functional (*Figure 3A*, *Figure 3—figure supplement 1*). Furthermore, the *V. radiata* CA1 subunit is also missing one of the three $Zn^{2+}$-coordinating histidine residues (Gln-135). Therefore, only one potentially catalytically active interface with all three $Zn^{2+}$ coordinating residues remains in *V. radiata*'s γCA—namely, the site between CA1 and CA2 at the "top" (most matrix-exposed periphery) of the domain. Clear density for a $Zn^{2+}$ can only be seen at this site (*Figure 3B*). In contrast, no $Zn^{2+}$ is seen at either of the two other sites, whose mutated residues are chemically incompatible with ion coordination. It is also important to note that the plant CA1, CA2 and CAL2 proteins belong to the CamH subclass of γCAs, which lack the acidic loop containing the catalytically important 'proton shuttle' glutamate

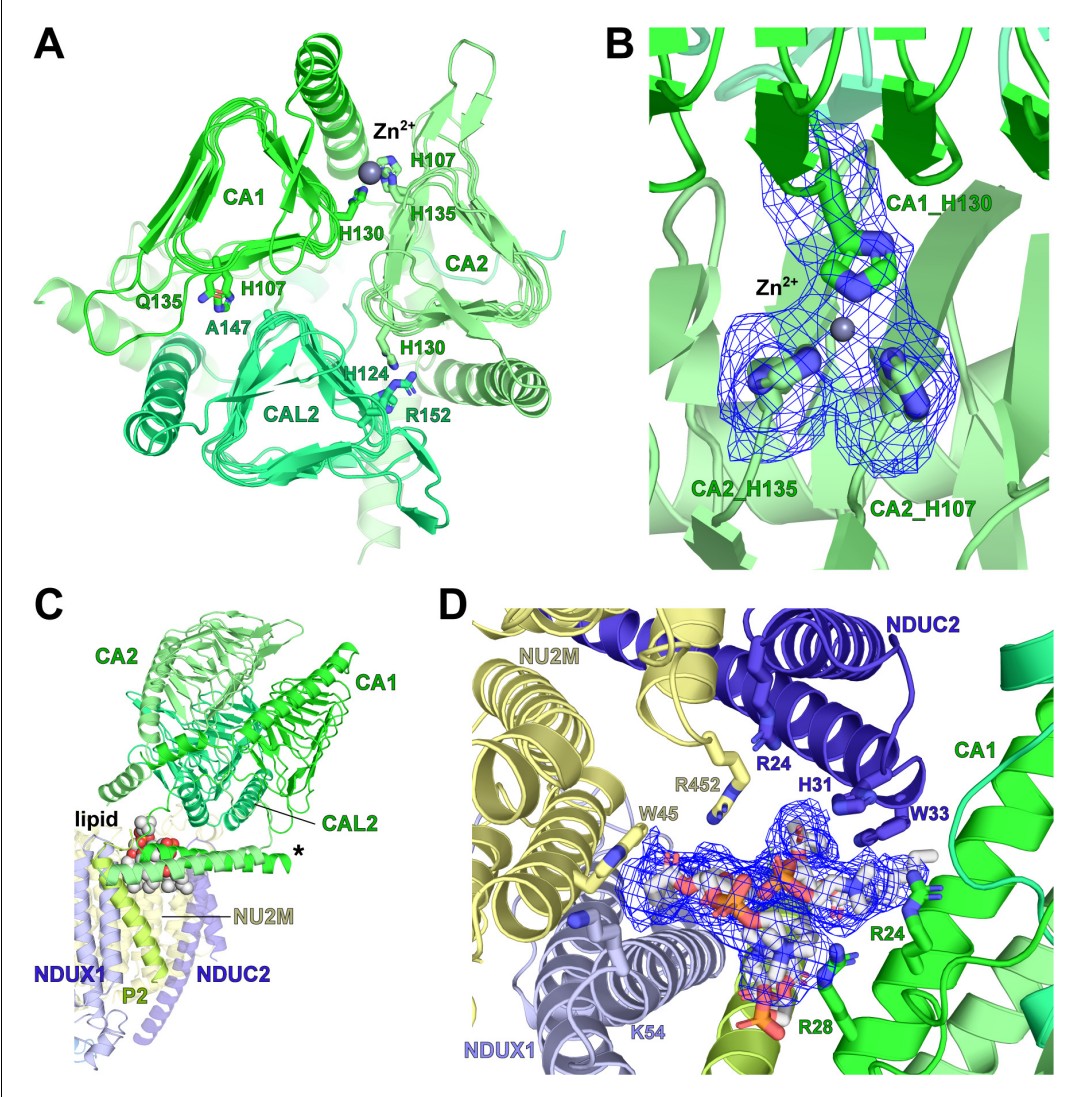

**Figure 3.** *V. radiata* γ-carbonic-anhydrase (γCA) domain, $Zn^{2+}$ coordination and associated lipid cavity. (**A**) Top view of the carbonic anhydrase domain with its CA1 (green), CA2 (lime) and CAL2 (lime green). Key residues at subunit interfaces for $Zn^{2+}$ coordination shown as sticks; $Zn^{2+}$ shown as grey sphere. Only the CA1-CA2 interface has all three key $Zn^{2+}$ coordinating histidines in place. (**B**) Zoom-in of $Zn^{2+}$ coordination site in (**A**), with map density for the three histidines and $Zn^{2+}$ shown as blue meshes. (**C**) Two phosphatidylcholines (spheres) are placed in the lipid cavity between the γCA and the $P_P$-module. Asterisk indicates the N-terminal amphipathic helices of CA1 and CA2. (**D** Zoom-in of the lipid cavity in **C**), with lipid density shown as blue mesh and key interacting residues shown as sticks.

The online version of this article includes the following figure supplement(s) for figure 3:

**Figure supplement 1.** Schematic representation of the γ-carbonic-anhydrase (γCA) domain interfaces and potential active sites in *V. radiata*.

residue (Glu89 in the canonical γCA from *Methanosarcina thermophila*) (*Zimmerman et al., 2010*). While some members of the CamH subclass are catalytically active, some are not (*Soto et al., 2006*; *Jeyakanthan et al., 2008*). Therefore, carbonic anhydrase activity of the γCA domain of CI must be confirmed experimentally (*Ferry, 2010*).

The other plant-specific subunit we were able to assign in CI* was the single-transmembrane subunit P2. This subunit binds on top of NDUX1, adjacent to NU2M and directly underneath the γCA domain. The N-terminus of P2 interacts directly with the γCA domain in the matrix. Together, P2, NDUX1, NU2M and NDUC2 form a lipid-filled cavity positioned directly below the γCA domain (*Figure 3C and D*). Several positively charged residues from the γCA domain subunits can be seen

**Table 5.** Quantification of interfaces within the γ-carbonic-anhydrase (γCA) domain and between γCA and the proximal pumping domain (P$_P$) of CI*.

Interface residues, surface areas, solvation free energies and P-values were determined by uploading the molecular model of CI* into the the PDBePISA tool for the exploration of macromolecular interfaces (*Krissinel and Henrick, 2007*). The table with the full list of interaction surfaces for CI* was filtered for the interfaces involving CA1, CA2 or CAL2. Total values were obtained by adding the relevant two-way interactions, as per PDBePISA guidelines.

| Subunit 1 | | | Subunit 2 | | | Inter-subunit interface | | |
|---|---|---|---|---|---|---|---|---|
| Subunit | # Interfacing residues | Interfacing surface area (Å$^2$) | Subunit | # Interfacing residues | Interfacing surface area (Å$^2$) | Interface surface area (Å$^2$) | Interface solvation free energy (kcal/mol) | Solvation free energy gain P-value |
| Within γCA domain | | | | | | | | |
| CA1 | 88 | 20,989 | CA2 | 80 | 22,088 | 2581.4 | −45.6 | 0.000 |
| CA1 | 79 | 20,989 | CAL2 | 72 | 20,073 | 2637.8 | −38.2 | 0.001 |
| CA2 | 73 | 22,088 | CAL2 | 66 | 20,073 | 2148.1 | −33.2 | 0.000 |
| | | | | | Total | 7367.3 | −117 | |
| Between γCA domain and membrane arm (P$_P$) | | | | | | | | |
| NU2M | 16 | 7610 | CA1 | 16 | 20,989 | 259.2 | −17.1 | 0.001 |
| NU2M | 16 | 7610 | CA2 | 16 | 22,088 | 261.1 | −17.1 | 0.001 |
| NU2M | 16 | 7610 | CAL2 | 16 | 20,073 | 263.6 | −18.0 | 0.003 |
| NDUC2 | 21 | 13,708 | CA1 | 21 | 20,989 | 427.8 | −20.2 | 0.000 |
| NDUC2 | 16 | 13,708 | CA2 | 16 | 22,088 | 242.3 | −15.8 | 0.000 |
| NDUC2 | 21 | 13,708 | CAL2 | 22 | 20,073 | 416.2 | −18.7 | 0.000 |
| P2 | 21 | 10,433 | CA1 | 21 | 20,989 | 362.6 | −16.5 | 0.000 |
| P2 | 26 | 10,433 | CA2 | 22 | 22,088 | 512.7 | −21.5 | 0.001 |
| P2 | 17 | 10,433 | CAL2 | 17 | 20,073 | 244.6 | −16.8 | 0.001 |
| NDUX1 | 16 | 13,955 | CA1 | 16 | 20,989 | 236.5 | −15.9 | 0.000 |
| NDUX1 | 17 | 13,955 | CA2 | 17 | 22,088 | 268.1 | −15.0 | 0.000 |
| NDUX1 | 16 | 13,955 | CAL2 | 16 | 20,073 | 240.9 | −16.7 | 0.000 |
| | | | | | Total | 3735.6 | −209.3 | |

interacting with these lipids, demonstrating that this lipid pocket also forms an important part of the γCA domain/membrane arm interface.

## Unassigned density

We were unable to assign four small regions of density in the CI* structure. One is the region near the N-terminus of NDUS8 discussed above (*Figure 4A*). Another is the likely C-terminal helix of NDUA6 also discussed above (*Figure 4B*). The third is on the intermembrane space side of the membrane arm (*Figure 4C*). In both *Y. lipolytica* and mammalian CI, this binding site is occupied by the C-terminus of the P$_P$- and P$_D$-module-spanning subunit NDUB5. In *Y. lipolytica* and mammals, NDUB5 spans nearly the entire length of the membrane arm. In *V. radiata* CI*, the density for this subunit follows the equivalent path of NDUB5 in *Y. lipolytica* and mammals but becomes disordered by the P$_P$-module's core subunit NU2M, which is adjacent to the C-terminus of accessory subunit NDUC2. The final stretch of unassigned density is for a single-transmembrane accessory subunit bound above NU6M TMH1 that contacts NU6M and NDUS5 on the intermembrane space side of the membrane arm (*Figure 4D*). This unassigned subunit protrudes away from CI* toward the location where CIII$_2$ binds in the mammalian supercomplex I+III$_2$ (*Letts and Sazanov, 2015*), suggesting a possible role for this subunit in supercomplex formation. No equivalent subunit is seen in either *Y. lipolytica* or mammalian CI, suggesting that this is a plant-specific subunit. However, due to local disorder, the density was too poor to assign the sequence from the reconstruction alone.

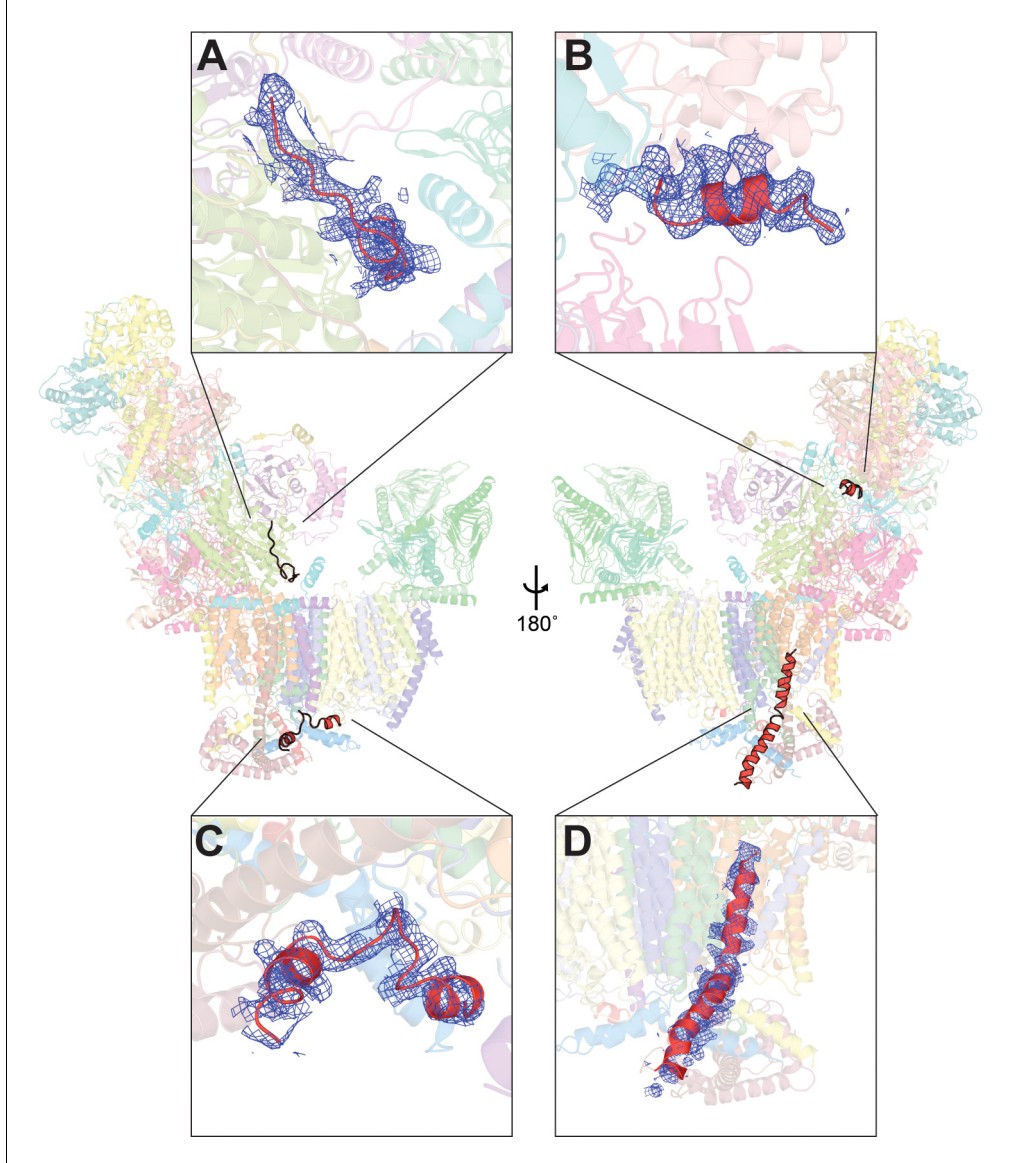

**Figure 4.** Unassigned density in *V. radiata* CI* map. Four stretches of unassigned, continuous densities in the map are shown with their positions on CI* indicated. Insets (**A-D**) show the density (blue mesh) and the poly-alanine chains (red) (**A, C, D**) or the putative NDUS6 C-terminal residues (**B**).

## Catalytic sites

All the cofactors necessary for the transfer of electrons between NADH and CoQ are present in the CI* intermediate. This includes the flavin mononucleotide (FMN) in NDUV1, all seven FeS-clusters of the main electron transport pathway (N3[V1], N1b[S1], N4[S1], N5[S1], N6a[S8], N6b[S8], N2[S7]), and the off-pathway FeS cluster N1a[V2] (*Figure 5A*). Moreover, density can be seen in the cryoEM map in the region of the Q-tunnel, in an equivalent position to that of CoQ in the *Y. lipolytica* structure (*Parey et al., 2019*; *Figure 5B*). This likely represents a CoQ molecule bound at the entry of the CI* Q-tunnel. However, this density is indistinct and thus we have not modeled a CoQ at this position. Analogously to the *Y. lipolytica* structure, no density for CoQ can be seen deeper in the Q-tunnel where CoQ would need to bind to accept electrons from the terminal FeS cluster (*Figure 5B*).

The loops that cap the Q-tunnel at the interface of the peripheral and membrane arms of the complex, namely the NU3M TMH1-2 and NU1M TMH5-6 loops, are disordered. This is analogous to

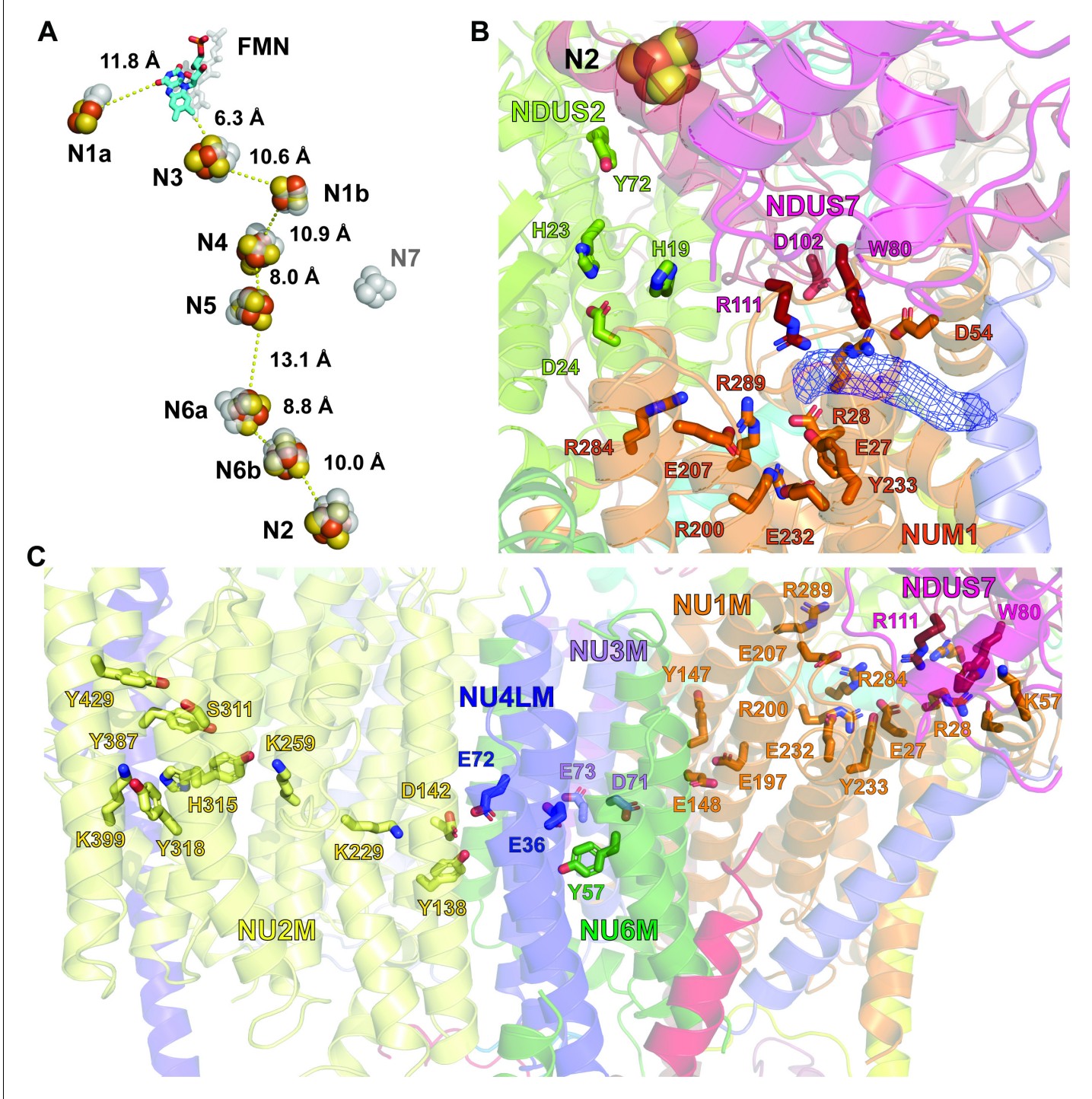

**Figure 5.** Structure of the redox centers, Q cavity and the hydrophilic axis of *V. radiata* CI*. (A) *V. radiata*'s FMN (stick) and iron-sulfur clusters (spheres) are labeled by nearest-atom center-to-center distances, overlaid with those from *T. thermophilus* (transparent grey). (B) Key residues (stick) delineating the Q cavity and the nearby N2 iron-sulfur cluster (spheres). Unassigned density in the Q cavity, potentially corresponding to quinone, shown as blue mesh. (C) Key CI* residues constituting the hydrophilic axis within the membrane domain shown as sticks.

what is observed in the open or deactive structures of the mammalian and *Y. lipolytica* complexes (*Agip et al., 2018*; *Letts et al., 2019*; *Parey et al., 2018*). Conformational changes in these loops are thought to play an important role in CI's coupling mechanism, which transduces the energy of NADH-quinone oxidoreduction in the Q module to proton pumping along the membrane arm

(*Parey et al., 2018*; *Cabrera-Orefice et al., 2018*). In particular, a π-bulge in NU6M's TMH3 in mammals has been seen to undergo a major conformational change, refolding into an α-helix during complex I's open-to-closed transition (*Agip et al., 2018*; *Letts et al., 2019*). This π-bulge in NU6M's TMH3 is also present in *V. radiata* CI*.

The 'E-channel' (*Baradaran et al., 2013*) and the hydrophilic axis of polar amino acid residues that are involved in proton translocation and span the membrane arm of CI are also evident in *V. radiata* CI* (*Figure 5C*). Given the lack of additional accessory subunits or assembly factors to cap the end of CI*'s shortened membrane arm, hydrophilic-axis residue Lys399 on NU2M's TMH12 is exposed to the midplane of the membrane. In all other structures of CI, the final transmembrane core subunit NU5M contains a transmembrane helix (TMH15) that caps the hydrophilic axis at the end of the transmembrane arm of full-length CI. The lack of such a cap on *V. radiata* NU2M in CI* suggests that, although Lys399 of NU2M is mostly surrounded by protein, the core hydrophilic axis may be in contact with lipid.

## Discussion

### Protein sample

The structure of *V. radiata* CI* presented here is the first atomic resolution structure of any plant mitochondrial electron transport chain complex and reveals several key features of mitochondrial CI from vascular plants.

CI* is an established assembly intermediate of plant CI, previously identified with genetic and proteomic studies in non-etiolated seedlings and mature leaves of *A. thaliana* and *N. sylvestris* (*Ligas et al., 2019*; *Meyer et al., 2011*; *Schertl et al., 2012*; *Schimmeyer et al., 2016*; *Senkler et al., 2017*; *Pineau et al., 2008*). Furthermore, CI* exhibits NADH-dehydrogenase activity in in-gel activity assays (*Meyer et al., 2011*; *Pineau et al., 2008*; *Haili et al., 2013*). Thus, it is unlikely that CI* in our mitochondrial preparations is a peculiarity of our etiolating growth conditions or our choice of model organism. Nevertheless, it may be the case that etiolating conditions promote the accumulation of CI* in *V. radiata* hypocotyls compared to seedlings grown in the light (see Appendix).

Moreover, it is also unlikely that CI* is a degradation product of CI rather than the assembly intermediate. Firstly, our membrane-extraction conditions (1% w:v digitonin, 4:1 g:g detergent:protein; see Materials and methods) are very gentle and were chosen after optimization to preserve protein: protein interactions in protein complexes and supercomplexes. Furthermore, immediately after extraction, we stabilize the detergent-extracted complexes with amphipathic polymers, which wrap around the complexes and further protect them from degradation/dissociation (*Breibeck and Rompel, 2019*). A large section of membrane stabilized and co-purified by our gentle digitonin/amphipol treatment is clearly seen around the perimeter of CI* at low contour (*Figure 1—figure supplement 4E*). Secondly, using digitonin at a higher concentration (5% w:v), an *A. thaliana* complexome profiling study (*Senkler et al., 2017*) obtained not only full-length CI and CI*, but also full-length CI in a higher order assembly with complex III (supercomplex SC I+III$_2$) [*Bultema et al., 2009*; *Dudkina et al., 2005*; *Eubel et al., 2004a*; *Eubel et al., 2004b*]. Protein:protein interactions between complexes in supercomplexes are known to be more labile than intra-complex protein:protein interactions. Given that the more fragile CI:CIII$_2$ interactions are maintained in 5% digitonin (*Senkler et al., 2017*), this argues that the presence of CI* —both in *Senkler et al., 2017* and in this study— is not due to a digitonin-induced dissociation of the P$_D$ domain, but rather that it is the true assembly intermediate. Thirdly, controlled-degradation experiments of plant CI in the presence of harsh detergents have shown that, analogous to mammalian CI, plant CI's detergent-induced dissociation occurs via detachment of the full peripheral arm (P$_P$-P$_D$) from the full matrix arm (N-Q) (*Klodmann et al., 2010*), *not* by dissociation between the P$_P$ and P$_D$ modules. Fourthly, we have reproducibly obtained the CI* fraction, which retains its in-gel and spectroscopic NADH-oxidase activity and chromatographic peak for several days, even after freeze/thaw cycles. For these reasons, it is evident that our structure corresponds to the CI* assembly intermediate, rather than to a degradation product of *V. radiata* CI.

## Carbonic anhydrase domain of plant CI

A major unique feature of plant CI compared to the other known structures is the large γCA domain located on the mitochondrial matrix side of the membrane arm of the complex (*Sunderhaus et al., 2006*).

Here, we were able to define the interface and anchoring interactions between the γCA domain and the rest of the complex at high resolution (*Figure 3*). In line with expectations from the early biochemical experiments on the plant γCA domain (*Sunderhaus et al., 2006*), the structure clearly shows that the interface between the γCA domain and the $P_P$-module is extensive and strong (*Table 5*). Additionally, we established that the γCA domain is membrane-targeted via two amphipathic helices that contact the CI membrane arm and through specific interactions with lipids in a lipid-filled pocket formed by core subunit NU2M, accessory subunits NDUX1, NDUC2 and plant-specific accessory subunit P2. Furthermore, our structure unambiguously resolves the identities of the hetero-trimeric components of the γCA domain of etiolated *V. radiata* as CA1, CA2 and CA2L. Unexpectedly, our structure also reveals that, due to this composition, only one out of the three potential active sites formed at the interfaces between CA1, CA2 and CA2L is capable of coordinating the $Zn^{2+}$ ion required for carbonic anhydrase catalysis. Nevertheless, whether the combination of γCA subunits and, consequently, the active site arrangements are different in different species, tissues or developmental stages (*Sunderhaus et al., 2006*; *Perales et al., 2004*; *Fromm et al., 2016*; *Ci˙Rdoba et al., 2019*) remains to be confirmed.

Structure alone is not sufficient to demonstrate catalytic ability of the plant CI γCA domain. Indeed, only bicarbonate binding to the plant mitochondrial γCAs has been shown (*Martin et al., 2009*) and, despite extensive attempts, no catalytic activity has been measured to date (*Fromm et al., 2016*; *Martin et al., 2009*). Further functional and structural studies with purified CI or CI* samples are necessary to determine whether the γCA domain possesses enzymatic activity.

## Structural insights on plant CI assembly

Less is known about CI assembly in plants than in fungi or metazoans (opisthokonts). In metazoans, detailed models of CI assembly have been generated and over a dozen CI assembly factors have been identified (*Formosa et al., 2018*; *Guerrero-Castillo et al., 2017*; *Garcia et al., 2017*). In plants, only three assembly factors have been thus far identified: L-galactono-1,4-lactone dehydrogenase (GLDH) (*Senkler et al., 2017*), the FeS protein INDH (*Wydro et al., 2013*) and an LYR protein termed CIAF1 (*Ivanova et al., 2019*). One possibility is that some of the unassigned densities observed in our reconstruction correspond to assembly factors that are bound to CI*. Current models of plant CI biogenesis predict that, of these three, only GLDH should be bound to the CI* intermediate (*Ligas et al., 2019*). However, GLDH is a large (~60 kDa) globular enzyme (*Leferink et al., 2008*), for which we do not see any consistent density in our structure. Nonetheless, it is possible that GLDH is bound via a flexible loop and thus averaged out in our reconstructions. Further assembly factors have been predicted to bind and cap NU2M in the membrane (*Ligas et al., 2019*). However, as noted above, we do not observe any additional transmembrane subunits capping the end of the shortened transmembrane arm.

There are major differences in CI assembly between plants and metazoans (*Figure 1—figure supplement 1*). In metazoans, the N-module (responsible for NADH oxidation) is assembled onto the Q-, $P_P$- and $P_D$-modules last (*Formosa et al., 2018*; *Guerrero-Castillo et al., 2017*; *Garcia et al., 2017*). This ensures that no assembly intermediate is capable of transferring electrons from NADH to CoQ. In contrast, in plants the final assembly step is the attachment of the $P_D$-module onto the CI* intermediate (*Ligas et al., 2019*). As noted above, the *V. radiata* CI* intermediate contains all of the subunits and co-factors needed to carry out NADH:CoQ oxidoreduction: CI* is, in principle, catalytically competent. Indeed, we were able to measure NADH-DQ oxidoreductase activity in the isolated CI* fraction (*Figure 1—figure supplement 2*).

The *V. radiata* CI* structure presented here reveals that this difference in assembly may in part stem from a significant difference in the structure of the peripheral-arm accessory subunit NDUS6. The plant NDUS6 subunit lacks an N-terminal domain relative to the NDUS6 homologues of opisthokonts. In opisthokonts, the N-terminal domain of NDUS6 binds over top of NDUA12 to interact with the Q-module accessory subunit NDUA9 (*Figure 2A*). Moreover, the assembly factor NDUFAF2 –a paralogue of NDUA12 that occupies the same binding site—sterically prevents the binding of

NDUS6 (*Parey et al., 2019*). Thus, in opisthokonts, NDUFAF2 must be removed and replaced with NDUA12 before NDUS6 can bind on the peripheral arm to complete the assembly of CI. In plants, a NDUFAF2 homologue on CI has yet to be observed experimentally (*Meyer et al., 2019*). Additionally, due to the lack of the N-terminal domain on NDUS6, plant NDUS6 does not cross over NDUA12 but binds next to it on the surface of the peripheral arm. Thus, in plants, NDUS6 may assemble on CI independent of the status of NDUFAF2/NDUA12. Furthermore, attaching the N-module before the $P_D$-module in plants may provide additional flexibility to their mitochondrial ETC (see discussion below and Appendix).

It is clear from the currently available structures that the interface between the $P_P$-module and $P_D$-module is more extensively stabilized by accessory subunit interactions in mammals than in *Y. lipolytica* or *V. radiata* (*Table 4*). Although we currently only have the structure of the CI* intermediate for *V. radiata* (which only contains the $P_P$-module), key truncations in core subunit NDUS2 and accessory subunits NDUA8 and NDUC2, discussed above (*Figure 2B and C*), already make this distinction clear. The lack of the NDUA8 and NDUC2 bridging interactions suggest that the interface between the $P_P$- and $P_D$-modules in plants may be weaker, which may also help explain the differences in the CI assembly pathway in plants versus opisthokonts. Identification of other possible bridging interactions across the $P_P$- and $P_D$-modules in plants will have to await the structure of full-length plant CI.

## Potential roles for CI* beyond CI assembly

The bioenergetic regulation of plants, which generate their energy through respiration and photosynthesis, is more intricate and dynamic than that of heterotrophs, whose main bioenergetic process is respiration. Mitochondrial respiration is the major source of ATP in plants' non-photosynthetic tissues such as roots. In photosynthetic tissue in the light, the role of mitochondrial respiration in ATP production is debated (*Shameer et al., 2019*; *Gardeström and Igamberdiev, 2016*) (see Appendix). Moreover, in photosynthetic tissue, conditions of intense light may lead to an over-production of reducing equivalents (NAD(P)H), which could be detrimental to the cells via the production of reactive oxygen species (ROS). To mitigate this, the plant mitochondrial electron transport chain (mETC) contains several 'alternative' oxidoreductases and oxidases that shunt electrons to molecular oxygen without pumping $H^+$, thus preventing the over-reduction of the NADH pool (*Millar et al., 2011*; *Schertl and Braun, 2014*). However, given that alternative complexes do not pump any $H^+$, energy is instead dissipated as heat.

Based on the fact that CI* is missing two of its four standard $H^+$ pumps (those in the $P_D$ module), and on our finding that CI* shows NADH-DQ oxidoreduction activity (*Figure 1—figure supplement 2*), we hypothesize that CI* may be an NADH-CoQ oxidoreductase with a lower $H^+$-pumping-to-electron-transfer ratio than full-length CI. Namely, we hypothesize that CI* could pump protons at a $2H^+:2e^-$ ratio rather than the $4H^+:2e^-$ of full-length CI (*Jones et al., 2017*).

Decreased $H^+:e^-$ ratios have previously been reported in functional yeast and bacterial CI mutants (*Dröse et al., 2011*; *Steimle et al., 2011*). A mutant of *Y. lipolytica* CI in which the $P_D$-module accessory subunit NB8M (homologue of plant NDUB7) is deleted (*nb8mΔ*) fails to assemble the $P_D$-module (*Dröse et al., 2011*). The resulting CI subcomplex is analogous to CI*, as it lacks only the $P_D$-module. The *nb8mΔ* mutant CI is a functional $H^+$-pumping NADH-CoQ oxidoreductase. However, its $H^+:e^-$ ratio, which is normally $4H^+:2e^-$ in fully assembled CI, is reduced to $2H^+:2e^-$ (*Dröse et al., 2011*). This is consistent with two of the four $H^+$-pumping subunits (NU4M and NU5M) being absent in the *nb8mΔ* mutant subcomplex. Similar results are seen in *E. coli* mutants with mutations in its distal $H^+$-pumping subunit NuoL (homologue of plant NU5M). Deletion of NuoL or truncation of its transmembrane helices 15–16, which bridge the $P_P$ and $P_D$ modules, result in a functional CI mutant whose $H^+:e^-$ coupling is $2H^+:2e^-$ (*Steimle et al., 2011*).

We hypothesize that a lower-$H^+$-pumping CI* could provide additional flexibility to plants' bioenergetic regulation, beyond the interplay between the canonical and alternative pathways of the mETC. For instance, having a $2H^+:2e^-$ ratio would allow CI* to contribute to ATP generation in situations where the mitochondrial $[NAD^+]/[NADH]$ ratio would not support $H^+$ pumping by CI (see Appendix for an in-depth discussion). Thus, CI* may provide additional energy-converting flexibility to plants' electron flow and energy conservation. This would be analogous to the flexibility seen for the electron transport chain of chloroplasts, which employ several dynamic mechanisms at different levels of regulation to adjust the $H^+:e^-$ coupling and the energetic and redox outputs to changing

environmental conditions (*Heber and Kirk, 1975*; *Scheibe et al., 2005*; *Rochaix, 2011*; *Murchie and Ruban, 2020*).

## Conclusion

Here, we present the structure of a mitochondria CI assembly intermediate, CI*, isolated from etiolated hypocotyls of *V. radiata*. CI* showed NADH-dehydrogenase activity in native in-gel and spectroscopic activity assays. Although we did not introduce experimental manipulations to prevent the assembly of mitochondrial CI, we were nonetheless able to isolate sufficient amounts of the CI* assembly intermediate for structure determination. This suggests that there are significant steady-state amounts of CI*in *V. radiata* mitochondria under these etiolating conditions and that CI* may be playing an independent physiological function beyond its role in CI assembly. The structure of *V. radiata* CI* presented here provides a wealth of information on mitochondrial CI composition, assembly and evolution and raises several questions on the dynamics and regulation of plant respiration. In order to address these questions, further research is needed into the structures of the fully assembled plant mitochondrial CI, as well as of its supercomplex with CIII$_2$. In addition, biochemical, cell biological and genetic approaches are paramount to test hypotheses on the potential functions of CI*.

# Materials and methods

**Key resources table**

| Reagent type (species) or resource | Designation | Source or reference | Identifiers | Additional information |
|---|---|---|---|---|
| Biological sample (*Vigna radiata*) | *V. radiata* seeds | Todd's Tactical Group | TS-229 | Lot SMU2-8HR; DOB 2/25/2019 |
| Commercial assay or kit | Pierce BCA assay kit | Thermo Fisher | 23225 | |
| Commercial assay or kit | 3–12% NativePAGE gels and buffers | Invitrogen | BN1001BOX; BN2001; BN2002 | |
| Chemical compound, drug | Digitonin, high purity | EMD Millipore | 300410 | |
| Chemical compound, drug | A8-35 | Anatrace | A835 | |
| Chemical compound, drug | Gamma-cyclodextrin | EMD Millipore | C4892 | |
| Chemical compound, drug | NADH | VWR Life Sciences | 97061–536 | |
| Chemical compound, drug | Nitrotetrazoleum | EMD Millipore | 74032 | |
| Software, algorithm | SerialEM | University of Colorado, *Schorb et al., 2019* | RRID:SCR_017293 | |
| Software, algorithm | RELION 3.0 | *Zivanov et al., 2018* | RRID:SCR_016274 | |
| Software, algorithm | Motioncor2 | *Zheng et al., 2017* | | |
| Software, algorithm | Ctffind4 | *Rohou and Grigorieff, 2015* | RRID:SCR_016732 | |
| Software, algorithm | crYOLO | *Wagner et al., 2019*; *Wagner and Raunser, 2020* | RRID:SCR_016732 | |

*Continued on next page*

*Continued*

| Reagent type (species) or resource | Designation | Source or reference | Identifiers | Additional information |
|---|---|---|---|---|
| Software, algorithm | Phyre2 | *Kelley et al., 2015* | | |
| Software, algorithm | Coot | *Emsley and Cowtan, 2004* | RRID:SCR_014222 | |
| Software, algorithm | PHENIX | *Liebschner et al., 2019*; *Goddard et al., 2018*; *Pettersen et al., 2004* | RRID:SCR_014224 | |
| Software, algorithm | UCSF Chimera | Resource for Biocomputing, Visualization, and Informatics at the University of California, San Francisco, *Pettersen et al., 2004* | RRID:SCR_004097 | |
| Software, algorithm | PyMOL Molecular Graphics System, Version 2.0 Schrödinger, LLC. | Schrödinger, LLC | RRID:SCR_000305 | Version 2.0 |
| Other | Holey carbon grids | Quantifoil | Q310CR1.3 | 1.2/1.3 300 mesh |

## *Vigna radiata* mitochondria purification

*V. radiata* seeds were purchased from Todd's Tactical Group (Las Vegas, NV). Seeds were incubated in 1% (v:v) bleach for 20 min and rinsed until the water achieved neutral pH. Seeds were subsequently imbibed in a 6 mM CaCl$_2$ solution for 20 hr in the dark. The following day, the imbibed seeds were sown in plastic trays on damp cheesecloth layers, at a density of 0.1 g/cm$^2$ and incubated in the dark at 20°C for 6 days. The resulting etiolated mung beans were manually picked, and the hypocotyls were separated from the roots and cotyledons. The hypocotyls were further processed for mitochondria purification based on established protocols (*Millar et al., 2007*). Briefly, hypocotyls were homogenized in a Waring blender with homogenization buffer (0.4 M sucrose, 1 mM EDTA, 25 mM MOPS-KOH, 10 mM tricine, 1% w:v PVP-40, freshly added 8 mM cysteine and 0.1% w:v BSA, pH 7.8) before a centrifugation of 10 min at 1000 x *g* (4°C). The supernatant was collected and centrifuged for 30 min at 12,000 x *g* (4°C). The resulting pellet was resuspended with wash buffer (0.4 M sucrose, 1 mM EDTA, 25 mM MOPS-KOH, freshly added 0.1% w:v BSA, pH 7.2) and gently centrifuged at 1000 x *g* for 5 min (4°C). This supernatant was then centrifuged for 45 min at 12,000 x *g*. The resulting pellet was resuspended in wash buffer, loaded on to sucrose step gradients (35% w:v, 55% w:v, 75% w:v) and centrifuged for 60 min at 52,900 x *g*. The sucrose gradients were fractionated with a BioComp Piston Gradient Fractionator (Fredericton, Canada) connected to a Gilson F203B fraction collector, following absorbance at 280 nm. The fractions containing mitochondria were pooled, diluted 1:5 in 10 mM MOPS-KOH, 1 mM EDTA, pH 7.2 and centrifuged for 20 min at 12,000 x *g* (4°C). The pellet was resuspended in final resuspension buffer (20 mM HEPES, 50 mM NaCl, 1 mM EDTA, 10% glycerol, pH 7.5) and centrifuged for 20 min at 16,000 x *g* (4°C). The supernatant was removed, and the pellets were frozen and stored in a −80°C freezer. The yield of these mitochondrial pellets was 0.8–1 mg per gram of hypocotyl.

## *Vigna radiata* mitochondrial membrane wash

Frozen *V. radiata* mitochondrial pellets were thawed at 4°C, resuspended in 10 ml of chilled (4°C) double-distilled water per gram of pellet and homogenized with a cold Dounce glass homogenizer. Chilled KCl was added to the homogenate to a final concentration of 0.15 M and further homogenized. The homogenate was centrifuged for 45 min at 32,000 x *g* (4°C). The pellets were resuspended in cold Buffer M (20 mM Tris, 50 mM NaCl, 1 mM EDTA, 2 mM DTT, 0.002% PMSF, 10% glycerol, pH 7.4) and further homogenized before centrifugation at 32,000 x *g* for 45 min (4°C). The pellets were resuspended in 3 ml of Buffer M per gram of starting material and further

homogenized. The protein concentration of the homogenate was determined using a Pierce BCA assay kit (Thermo Fisher, Waltham, MA), and the concentration was adjusted to a final concentration of 10 mg/ml and 30% glycerol.

## Extraction and purification of mitochondrial complexes

Washed membranes were thawed at 4°C. Digitonin (EMD Millipore, Burlington, MA) was added to the membranes at a final concentration of 1% (w:v) and a digitonin:protein ratio of 4:1. Membranes complexes were extracted by tumbling this mixture for 60 min at 4°C. The extract was centrifuged at 16,000 x *g* for 45 min (4°C). Amphipol A8-35 (Anatrace, Maumee, OH) was added to the supernatant at a final concentration of 0.2% w:v and tumbled for 30 min at 4°C, after which gamma-cyclodextrin (EMD Millipore, Burlington, MA) was added to a final amount of 1.2x gamma-cyclodextrain:digitonin (mole:mole). The mixture was centrifuged at 137,000 x *g* for 60 min (4°C). The supernatant was concentrated with centrifugal protein concentrators (Pall Corporation, NY, NY) of 100,000 MW cut-off, loaded onto 10–45% (w:v) or 15–45% (w:v) linear sucrose gradients in 15 mM HEPES, 20 mM KCl, pH 7.8 produced using factory settings of a BioComp Instruments (Fredericton, Canada) gradient maker and centrifuged for 16 hr at 37,000 x *g* (4°C). The gradients were subsequently fractionated with BioComp Piston Fractionatr connected to a Gilson F203B fraction collector, following absorbance at 280 nm. Select fractions were pooled, concentrated with protein concentrators (Pall Corporation, NY, NY) of 100,000 MW cut-off and purified on a Superose6 10/300 chromatography column (GE Healthcare, Chicago, IL) using an NGC 10 Medium-Pressure chromatography system (Biorad, Hercules, CA). For grid preparation, the relevant fractions were buffer-exchanged into 20 mM HEPES, 150 mM NaCl, 1 mM EDTA, pH 7.8 (no sucrose) and concentrated to a final protein concentration of 6 mg/ml and mixed one-to-one with the same buffer containing 0.2% digitonin (w:v),for a final concentration of 0.1% digitonin (w:v).

## BN-PAGE

Mitochondrial membrane extractions were diluted in 2X BN-loading buffer (250 mM aminocaproic acid, 100 mM Tris-HCl, pH 7.4, 50% glycerol, 2.5% (w:v) Coomassie G-250), loaded on pre-cast 3–12% NativePAGE Bis-Tris gels (Invitrogen, Carlsbad, CA) and run at 4°C. The cathode buffer was 50 mM Tricine, 50 mM BisTris-HCl, pH 6.8 plus 1X NativePAGE Cathode Buffer Additive (0.02% Coomassie G-250) (Invitrogen, Carlsbad, CA) and the anode buffer was 50 mM Tricine, 50 mM BisTris-HCl, pH 6.8. Gels were run at 200 V constant voltage for ~30 min, after which the cathode buffer was switched for a 'light blue' cathode buffer containing 50 mM Tricine, 50 mM BisTris-HCl, pH 6.8 plus 0.1X NativePAGE Cathode Buffer Additive (0.002% Coomassie G-250) (Invitrogen, Carlsbad, CA). The settings were changed to 7 mA constant amperage and run for another ~90 min.

## Activity assays

The CI in-gel NADH dehydrogenase activity assay was performed based on *Schertl and Braun, 2015*. The BN-PAGE gel was incubated in 10 ml of freshly prepared reaction buffer (1 mg/ml nitrotetrazoleum blue in 10 mM Tris-HCl pH 7.4). Freshly thawed NADH was added to the container with the gel, to a final concentration of 150 µM. The gel with the complete reaction buffer was rocked at room temperature for ~10 min. Once purple bands indicating NADH-dehydrogenase activity appeared, the reaction was quenched with a solution of 50% methanol (v:v) and 10% acetic acid (v: v).

The spectroscopic NADH dehydrogenase activity assay was performed based on *Huang et al., 2015*; *Letts et al., 2019*. CI NADH:decylubiquinone (DQ) activity was measured by spectroscopic observation of NADH oxidation at 340 nm wavelength at 30°C using a Molecular Devices (San Jose, CA) Spectramax M2 spectrophotometer. Reactions were carried out in 96-well plates. Protein samples were added to 190 µL of reaction buffer (100 mM HEPES, pH 7.4, 50 mM NaCl, 10% glycerol, 4 µM KCN, 1 mg/ml BSA, 10 µM cyt *c*, with or without 100 µM DQ as required) and mixed by pipetting. The reaction was initiated by addition of NADH to a final concentration of 150 µM and briefly mixed by pipetting and plate stirring for 10 s before recording. Measurements were done in triplicate, averaged and background-corrected. The known extinction co-efficient of NADH (6.22 mM$^{-1}$ cm$^{-1}$) was used in the calculations. Statistical significance was determined using a two-tailed t-test.

## CryoEM data acquisition

The CI* sample (6 mg/ml protein in 20 mM HEPES, 150 mM NaCl, 1 mM EDTA, 0.1% digitonin, pH 7.8) was applied onto glow-discharged holey carbon grids (Quantifoil, 1.2/1.3 300 mesh) followed by a 60 s incubation and blotting for 9 s at 15°C with 100% humidity and flash-freezing in liquid ethane using a FEI Vitrobot Mach III.

CryoEM data acquisition was performed on a 300 kV Titan Krios electron microscope equipped with an energy filter and a K3 detector at the UCSF W.M. Keck Foundation Advanced Microscopy Laboratory, accessed through the Bay Area Cryo-EM Consortium. Automated data collection was performed with the SerialEM package (*Schorb et al., 2019*). Micrographs were recorded at a nominal magnification of 60,010 X, resulting in a pixel size of 0.8332 $Å^2$. Defocus values varied from 1.5 to 3.0 µm. The dose rate was 20 electrons per pixel per second. Exposures of 3 s were dose-fractionated into 118 frames, leading to a dose of 0.72 electrons per $Å^2$ per frame and a total accumulated dose of 86.4 electrons per $Å^2$. A total of 9816 micrographs were collected, 8541 of which were used for further analysis.

## Data processing

Software used in the project was installed and configured by SBGrid (*Morin et al., 2013*). All processing steps were done using RELION 3.0 (*Zivanov et al., 2018*) unless otherwise stated. Motioncor2 (*Zheng et al., 2017*) was used for whole-image drift correction of each micrograph. Contrast transfer function (CTF) parameters of the corrected micrographs were estimated using Ctffind4 (*Rohou and Grigorieff, 2015*) and refined locally for each particle in RELION. Automated particle picking using crYOLO (*Wagner et al., 2019*; *Wagner and Raunser, 2020*) resulted in ~1.5 million particles. The particles were extracted using $400^2$ pixel box binned two-fold and sorted by reference-free 2D classification followed by re-extraction at $512^2$ pixel box. Reference-free 2D classification resulted in the identification of 190,951 CI* particles. An *ab initio* model was generated in RELION from these particles (*Punjani et al., 2017*). This model, lowpass-filtered at 30 Å, was used for initial 3D classification with a regularization parameter T of 4. This initial processing resulted in ~34,000 particles of good quality, which separated into a single class (*Figure 1—figure supplement 3C*). The best class was refined to a nominal resolution of 3.9 Å according to the gold standard FSC criteria (*Scheres and Chen, 2012*). It was clear that the local resolution of this refinement was impacted by hinge-like motions between the membrane and peripheral arms of the complex. Therefore, sub-region refinements were also performed masking around the membrane arm and peripheral arm, respectively (*Figure 1—figure supplement 3C*). This resulted in significantly, improved map quality, especially for the γCA domain on the membrane arm (*Figure 1—figure supplement 3C*). These improved maps were used for model building and refinement. The two focused refined maps were then combined into a composite map using Phenix.

## Model building and refinement

Starting models for isolated ovine CI (*Letts and Sazanov, 2015*) and bacterial γCA (*Iverson et al., 2000*), corrected for the *V. radiata* sequence, were used as templates. Additionally, starting models were generated using the Phyre2 web portal (*Kelley et al., 2015*). These models were split and fit into the highest-resolution focused refinement maps for separate atomic model building of the CI* peripheral arm and CI* membrane arm in Coot (*Emsley and Cowtan, 2004*). Real-space refinement of the model was done in PHENIX (*Liebschner et al., 2019*; *Goddard et al., 2018*; *Pettersen et al., 2004*) and group atomic displacement parameters (ADPs) were refined in reciprocal space. The single cycle of group ADP refinement was followed by three cycles of global minimization, followed by an additional cycle of group ADP refinement and finally three cycles of global minimization (*Letts et al., 2019*).

## Model interpretation and figure preparation

Molecular graphics and analyses were performed with UCSF Chimera (*Pettersen et al., 2004*), developed by the Resource for Biocomputing, Visualization, and Informatics at the University of California, San Francisco, with support from NIH P41-GM103311, as well as the PyMOL Molecular Graphics System, Version 2.0 Schrödinger, LLC.

## Acknowledgements
We are grateful to K Abe, MG Zaragoza, C Goodwin, R Murguia and C Bower for help with *V. radiata* growth and mitochondrial isolations. Data was collected at the UCSF WM Keck Foundation Advanced Microscopy Laboratory, accessed through the Bay Area Cryo-EM Consortium BACEM, with the assistance of D Bulkley and Z Yu. We are grateful to J Al-Bassam, E Baldwin, JC Lagarias and J Callis for helpful input on the manuscript and to W Broadly of the UC Davis High-Performance Cluster for technical assistance. MM acknowledges funding from the UC Davis POP Program.

## Additional information

### Funding
No external funding was received for this work.

### Author contributions
Maria Maldonado, Conceptualization, Formal analysis, Supervision, Investigation, Visualization, Writing - original draft, Project administration, Writing - review and editing, Validation and Funding acquisition; Abhilash Padavannil, Long Zhou, Formal analysis, Validation, Investigation, Visualization; Fei Guo, Validation, Investigation; James A Letts, Conceptualization, Data curation, Formal analysis, Supervision, Funding acquisition, Validation, Visualization, Writing - original draft, Project administration, Writing - review and editing

### Author ORCIDs
Maria Maldonado (iD) https://orcid.org/0000-0002-3428-1053
James A Letts (iD) https://orcid.org/0000-0002-9864-3586

### Decision letter and Author response
Decision letter https://doi.org/10.7554/eLife.56664.sa1
Author response https://doi.org/10.7554/eLife.56664.sa2

## Additional files

### Supplementary files
• Transparent reporting form

### Data availability
Deposition of structural models and maps to the PDB and EMDB are: EMD-22090, PDB ID 6X89, EMD-22091, EMD-22092, EMD-22093.

The following datasets were generated:

| Author(s) | Year | Dataset title | Dataset URL | Database and Identifier |
|---|---|---|---|---|
| Maldonado M, Padavannil A, Zhou L, Letts JA | 2020 | *Vigna radiata* mitochondrial complex I* | http://www.rcsb.org/structure/6X89 | RCSB Protein Data Bank, 6X89 |
| Maldonado M, Guo F, Letts JA | 2020 | *Vigna radiata* mitochondrial complex I* | http://www.ebi.ac.uk/pdbe/entry/emdb/EMD-22090 | Electron Microscopy Data Bank, EMD-22090 |
| Maldonado M, Guo F, Letts JA | 2020 | *Vigna radiata* mitochondrial complex I* | http://www.ebi.ac.uk/pdbe/entry/emdb/EMD-22091 | Electron Microscopy Data Bank, EMD-22091 |
| Maldonado M, Guo F, Letts JA | 2020 | *Vigna radiata* complex I* membrane arm. | http://www.ebi.ac.uk/pdbe/entry/emdb/EMD-22092 | Electron Microscopy Data Bank, EMD-22092 |
| Maldonado M, Guo F, Letts JA | 2020 | *Vigna radiata* complex I* peripheral arm | http://www.ebi.ac.uk/pdbe/entry/emdb/EMD-22093 | Electron Microscopy Data Bank, EMD-22093 |

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

# Appendix 1

## Bioenergetic Considerations of CI* H$^+$ pumping

### 1.Background

The purpose of this Appendix is to discuss the bioenergetic implications of CI*'s potential function as a $2e^-$:$2H^+$ redox-coupled proton pump and what, if any, possible roles it could play in plant bioenergetic physiology.

For plant cells, there is much debate about the roles of the various pathways and organelles in supplying energy to the cytoplasm and reduction equivalents to the peroxisome (*O'Leary et al., 2019*; *Gardeström and Igamberdiev, 2016*). Energy is provided to the cytoplasm and peroxisome in the form of ATP and/or reduction equivalents (NAD(P)H), from the chloroplasts and mitochondria through many pathways. Pathways that export ATP or reduction equivalents from the chloroplast to the cytoplasm include: 1) the chloroplast malate valve; 2) the triose phosphate-3-phosphglycerate (TP-2PGA) shuttle and 3) the phosphoenolpyruvate (PEP)-pyruvate shuttle. ATP can also be imported into the chloroplast from the cytosol via the plastidial nucleotide translocase to energize the chloroplast at night (*Flügge et al., 2011*). Pathways that export ATP or reduction equivalents directly from the mitochondria to the cytoplasm include: 1) the mitochondrial malate valve; 2) the mitochondrial adenylate nucleotide translocase.

Furthermore, photosynthesis and mitochondrial respiration are linked *via* the photorespiratory C$_2$ cycle and the peroxisome (*Husic et al., 1987*). In this pathway, 2-phosphoglycerate generated by the oxygenase activity of RuBisCO is converted to glycolate and transported into the peroxisome, where it is converted into glycine. Glycine is then transported into the mitochondria, where it is converted by the glycine decarboxylase complex (GDC) into serine, CO$_2$ and NH$_3^+$, reducing NAD$^+$ to NADH in the mitochondrial matrix. These reduction equivalents can then be exported to the cytoplasm by the mitochondrial malate valve or be fed into oxidative phosphorylation *via* complex I (CI). In order to complete the photorespiratory pathway, serine must be transported back to the peroxisome, where is it reduced to glycerate, which is then transported back into the chloroplast. The reduction equivalents for the conversion of serine to glycerate in the peroxisome are provided from the cytoplasm by the malate-oxaloacetate shuttle of the peroxisome.

Recent detailed modelling of the energetic coupling between organelles in plant cells revealed that, although chloroplasts can theoretically generate sufficient ATP to satisfy the energy requirements of the entire plant cell, this would require unrealistic light-use efficiency and higher-than-available levels of ATP export from the chloroplast (*Shameer et al., 2019*). Although still controversial, these modeling results in conjunction with some experimental results (reviewed in *Gardeström and Igamberdiev, 2016*) suggest that during photosynthesis the bulk of *cytosolic* ATP is provided by the mitochondria. Furthermore, these studies suggest that rather than being exported for use by the peroxisome in photorespiration, the reduction equivalents (NADH) generated by GDC in the mitochondrial matrix are used directly for ATP production by the mitochondrial electron transport chain (mETC) and the needs of the peroxisome are mainly met by chloroplast-derived reduction equivalents (*Shameer et al., 2019*). Moreover, it is important to note that the photorespiratory C$_2$ cycle is a high-flux pathway in C$_3$ plants such as *V. radiata*, with a flux approximately equal on a molar basis to the flux through the photosynthetic C$_3$ cycle (*Oliver, 1994*). The total CO$_2$ released by GDC is ~25% of the moles of C fixed by RuBisCO (*Husic et al., 1987*; *Oliver, 1994*).

Given the above considerations, it is reasonable to conclude that a major source of NADH for mitochondrial respiration in actively photosynthesizing cells is generated by GDC and that these levels may fluctuate as a function of photosynthetic output, driven by light availability.

In this Appendix, we consider the effect of the fluctuations of the mitochondrial-matrix [NAD+]/[NADH] ratio on the reaction catalyzed by CI. We then discuss how alternative metabolic routes in the mETC, such as alternative NADH dehydrogenases (NDs), the alternative oxidase and possibly CI*, may provide plants with respiratory flexibility to manage a range of different and changing conditions (see also the review by *O'Leary et al., 2019*).

### 2.Complex I reaction

Mitochondrial CI catalyzes the reaction:

$$NADH + CoQ + 2H_N^+ + n_p H_N^+ \rightleftharpoons NAD^+ + H_N^+ + CoQH_2 + n_p H_P^+$$
$$\text{Reaction 1}$$

Where:

- NADH is the reduced form of nicotinamide adenine dinucleotide
- CoQ is the oxidized form of coenzyme Q (ubiquinone)
- $H_N^+$ represents a proton on the negative (N) side of the membrane (mitochondrial matrix)
- $n_p$ is the number of H$^+$ pumped across the inner mitochondrial membrane by CI
- NAD$^+$ is the oxidized from of nicotinamide adenine dinucleotide
- CoQH$_2$ is the reduced from of coenzyme Q (ubiquinol)
- $H_P^+$ represents a proton on the positive (P) side of the membrane (inter-membrane space)

The Gibbs energy change ($\Delta$G$^{CI}$) of the CI reaction can be determined by splitting the reaction into its separate electron transfer and H$^+$-pumping parts. For completeness, we will briefly derive the expression for these two parts here and then combine them into the final expression for $\Delta$G$^{CI}$.

## 2.1 Electron Transfer

The above oxidoreduction reactions for NADH and CoQ can be represented by two half reactions:

$$NAD^+ + H_N^+ + 2e^- \rightleftharpoons NADH$$
$$\text{Reaction 2}$$

$$CoQ + 2H_N^+ + 2e^- \rightleftharpoons CoQH_2$$
$$\text{Reaction 3}$$

The midpoint potential at which the concentrations of the reduced and oxidized forms are equal at pH 7.0 ($E_{m,7}$) for these half reactions are known to be -320 mV for Reaction 2 and 4 mV for Reaction 3 (see *Appendix 1—table 1* for references).

**Appendix 1—table 1.** Values used in the calculations.

| Variable | Value | Source |
|---|---|---|
| $\Delta$p | 160 mV | Values of 140–190 mV have been reported from respiring cells (*Ripple et al., 2013*); a value of 200 mV was reported for isolated etiolated *V. radiata* mitochondria after addition of 1 mM NADH, which defines an upper limit for steady-state respiration (*Moore and Bonner, 1981*) |
| R | 8.314 kJ K$^{-1}$ mol$^{-1}$ | Physical Constant |
| T | 300 K | Approximately 27 ˚C |
| F | 96,485 C mol$^{-1}$ | Physical Constant |
| $E_{m,7}^{CoQ}$ | 4 mV | This value varies as a function of pH so should only be considered an estimate (*Nicholls, 2013*) |
| $[CoQ]_{IMM} / [CoQH_2]_{IMM}$ | 10 | *Kim et al., 2012* |
| $E_{h,7}^{CoQ}$ | 34 mV | Calculated from $E_{m,7}^{CoQ}$ and $[CoQ]_{IMM}/[CoQH_2]_{IMM}$ using *Equation 1* |
| $E_{m,7}^{NADH}$ | −320 mV | This value varies as a function of pH, so should only be considered an estimate (*Nicholls, 2013*) |

The redox potential of the half reactions at pH 7 can be calculated using the following equation:

$$E_{h,7} = E_{m,7} + \frac{2.3RT}{nF}\log_{10}\left(\frac{[oxidised]}{[reduced]}\right) \quad (1)$$

Where:

- $E_{h,7}$ is the redox potential

- $E_{m,7}$ is the midpoint potential
- $R$ is the gas constant (8.314 kJ K$^{-1}$ mol$^{-1}$)
- The factor of 2.3 originates from converting the natural logarithm to log$_{10}$
- $T$ is the absolute temperature (K)
- $n$ is the number of electrons transferred in the half reaction
- $F$ is Faraday's constant (96,485 C mol$^{-1}$)
- [oxidized] is the actual concentration of the oxidized form
- [reduced] is the actual concentration of the reduced form

The redox potential difference between the NADH and CoQ pools is defined as the difference in their redox potential:

$$\Delta E_h = E_{h,7}^{CoQ} - E_{h,7}^{NADH} \tag{2}$$

Where:

- $\Delta E_h$ is the redox potential difference
- $E_{h,7}^{UQ}$ is the redox potential for CoQ
- $E_{h,7}^{NADH}$ is the redox potential for NADH

$\Delta E_h$ as presented in **Equation 2** is also known as the redox span of CI ($\Delta E_s^{CI}$). The redox span of CI is related to the Gibbs energy change accompanying the electron transfer ($\Delta G_{ET}$) between the couples by:

$$\Delta G_{ET} = -2F\Delta E_s^{CI} \tag{3}$$

Where:

- $\Delta G_{ET}$ is the Gibbs energy change of the electron transfer
- 2 is the number of electrons transferred
- $F$ is Faraday's constant (96,485 C mol$^{-1}$)

## 2.2 Proton pumping

In the general case for the Gibbs energy change ($\Delta G$) accompanying the transport of an ion across a membrane, the ion will be affected by both concentrative and electrical gradients:

$$\Delta G = -mF\Delta\Psi + RT\ln\left(\frac{[X^{m+}]_P}{[X^{m+}]_N}\right) \tag{4}$$

Where:

- $m$ is the charge of the ion
- $F$ is Faraday's constant (96,485 C mol$^{-1}$)
- $\Delta\Psi$ is the membrane potential
- $R$ is the gas constant (8.314 kJ K$^{-1}$ mol$^{-1}$)
- $T$ is the absolute temperature (K)
- $[X^{m+}]_P$ is the concentration of ions on the P side of the membrane
- $[X^{m+}]_N$ is the concentration of ions on the N side of the membrane

This is often expressed as the ion electrochemical gradient $\Delta\tilde{\mu}_X^{m+}$ with units of kJ mol$^{-1}$. For a proton electrochemical gradient $\Delta\tilde{\mu}_{H^+}$, **Equation 4** can be simplified as pH is a logarithmic function of [H$^+$]:

$$\Delta\tilde{\mu}_{H^+} = -F\Delta\Psi + 2.3RT\Delta\text{pH} \tag{5}$$

Where:

- $\Delta$pH is defined as the pH on the P side of the membrane minus the pH on the N side (pH$_P$-pH$_N$)
- The factor of 2.3 comes from converting the natural logarithm to log$_{10}$

The proton motive force (PMF or Δp) was defined by Peter Mitchell to convert $\Delta\tilde{\mu}_{H^+}$ into units of voltage to facilitate comparison with redox potential differences:

$$\Delta p = -\frac{\Delta\tilde{\mu}_{H^+}}{F} \tag{6}$$

Where:

- $F$ is Faraday's constant (96,485 C mol$^{-1}$)

## 2.3 Combining terms for overall ΔG expression

Given *Equations 3 and 5*, the overall ΔG of Reaction 1 catalyzed by CI can be given as:

$$\Delta G^{CI} = -2F\Delta E_s^{CI} + n_p\Delta\tilde{\mu}_{H^+} \tag{7}$$

Using *Equation 6* we obtain:

$$\Delta G^{CI} = F\left(n_p\Delta p - 2\Delta E_s^{CI}\right) \tag{8}$$

Using *Equation 2* we obtain:

$$\Delta G^{CI} = F\left(n_p\Delta p - 2\left(E_{h,7}^{CoQ} - E_{h,7}^{NADH}\right)\right) \tag{9}$$

Finally, using *Equation 1* we obtain:

$$\Delta G^{CI} = F\left(n_p\Delta p - 2\left(E_{h,7}^{CoQ} - \left(E_{m,7}^{NADH} + \frac{2.3RT}{nF}\log_{10}\left(\frac{[NAD^+]_M}{[NADH]_M}\right)\right)\right)\right) \tag{10}$$

Equation 10 allows us to express the Gibbs energy change for CI at a given proton motive force (Δp) and redox poise of the CoQ pool $\left(E_{h,7}^{CoQ}\right)$ as a function of the number of H$^+$ pumped $(n_p)$ and the ratio of NAD$^+$ to NADH in the mitochondrial matrix $\left(\frac{[NAD^+]_M}{[NADH]_M}\right)$ (*Appendix 1—table 1*, *Appendix 1—figure 1*).

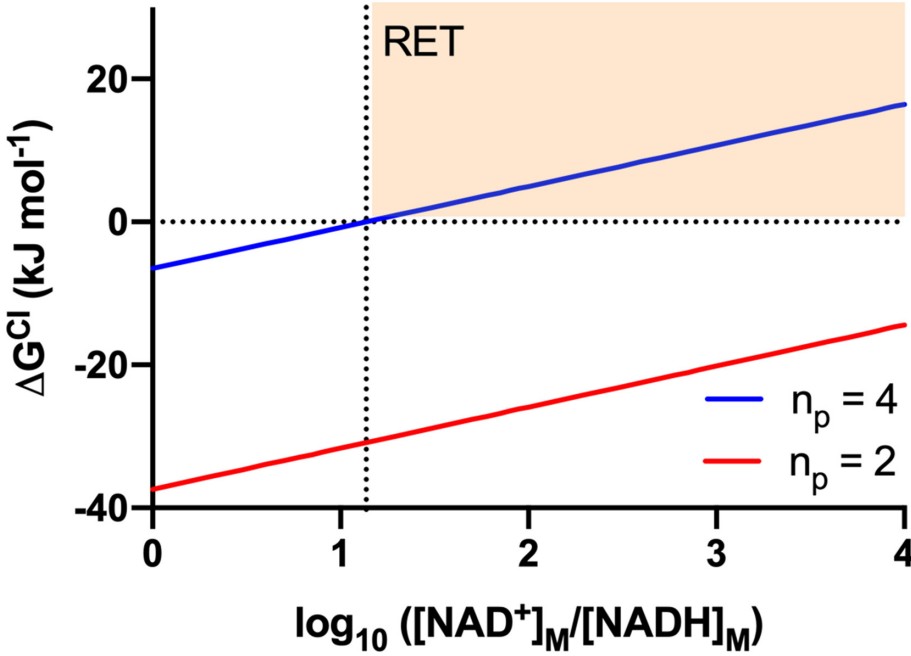

**Appendix 1—figure 1.** The Gibbs energy change of the CI reaction (ΔG$^{CI}$) as a function of the redox poise of the mitochondrial NADH pool. The Gibbs energy change was calculated using equation 10 and the values presented in Table A1, for reactions in which CI pumps 4 H$^+$ (blue; representative of

the standard, full-length CI pumping with a 4H$^+$:2e$^-$ ratio) or 2 H$^+$ (red; representative of a putative CI* pumping with a 2H$^+$/2e$^-$ ratio). The horizontal dashed line indicates equilibrium state ($\Delta G^{CI} = 0$) for the different [NAD$^+$]/[NADH] ratios. The vertical dashed line indicates the [NAD$^+$]/[NADH] ratio at which full-length CI (blue) attains equilibrium ($\epsilon = 1$). The highlighted orange region corresponds to conditions in which thermodynamics would favor reverse electron transport (RET) by full-length CI ($\epsilon > 1$).

At negative values of $\Delta G^{CI}$, the CI reaction occurs in the forward direction, that is, NADH oxidation and proton pumping into the intermembrane space. At positive values of $\Delta G^{CI}$, the CI reaction occurs in the reverse direction, usually associated with large amounts of reactive oxygen species (ROS) production, which are known to be detrimental to the cell (*Robb et al., 2018*). This reverse reaction is also called 'reverse electron transport' (RET).

## 2.4. Thermodynamic efficiency

The thermodynamic efficiency ($\epsilon$) of the reaction is defined as the fraction of the energy released on electron transfer that is transduced into the proton motive force:

$$\epsilon = \frac{n_p \Delta \mathrm{p}}{2\Delta E_S^{CI}} \tag{11}$$

By definition, $\epsilon < 1$ if the CI reaction is in the forward direction (i.e., oxidation of NADH), $\epsilon = 1$ at equilibrium (where $n_p \Delta \mathrm{p} = 2\Delta E_S^{CI}$) and $\epsilon > 1$ during RET by CI (i.e. ROS production).

## 3.Discussion

Although the energetic analysis presented here does not take into account the dynamics of the electron transport system (i.e., fluctuations in $\Delta \mathrm{p}$ and $E_{m,7}^{CoQ}$ caused by fluctuations in [NAD$^+$]/[NADH]), several important conclusions can be drawn. (For clarity, we call a CI pumping with a 4H$^+$:2e$^-$ratio simply 'CI'. For the sake of argument, we assume that CI* is a CI entity that pumps protons with a 2H$^+$:2e$^-$ ratio.):

1. When CI is operating near equilibrium, CI* runs irreversibly in the forward direction. CI* would be incapable of operating in ROS-generating RET mode in a range of [NAD+]/[NADH] up to values of ~10$^5$.
2. Under conditions that favor RET from CI (e.g. a drop in [NADH], leading to an increased [NAD+]/[NADH] ratio), CI* continues to work in the forward direction. This could help maintain the inner mitochondrial membrane proton motive force ($\Delta \mathrm{p}$) at high [NAD+]/[NADH].
3. CI has a higher thermodynamic efficiency than CI* under all conditions that favor the forward direction for CI ($\Delta G^{CI} < 0$). Nevertheless, under these conditions, CI* would still transduce energy from NADH into $\Delta \mathrm{p}$. Due to the factor of 2 difference in H$^+$-pumping, CI* would always transduce half the amount of energy compared to CI (see *Equation 11*). For example, under conditions of near equilibrium for CI ($\epsilon \approx 1$), CI* would still transduce energy at 50% efficiency ($\epsilon \approx 0.5$).

From this analysis, we conclude that one of the key advantages of having a CI* pumping at 2H$^+$:2e$^-$ would be that it could still work in the forward direction, maintaining the inner mitochondrial membrane's proton motive force, in situations where full-length CI would operate in the reverse direction and generate ROS. However, in order for plants to take full advantage of this potential bioenergetic benefit of partial energy transduction by CI*, plant CI should display a strong rectification that prevents RET. In other words, there should be a mechanism that strongly inhibits CI turnover in conditions where it would otherwise run in reverse and lead to oxidative damage.

This type of rectification of CI has been proposed to exist in mammals *via* CI's active-to-deactive transition as a way to prevent oxidative damage upon ischemic reperfusion (*Chouchani et al., 2013*). The active-to-deactive transition of mitochondrial CI has been observed in several but not all studied fungi and metazoans; moreover, it is absent in all prokaryotic CI thus far examined (reviewed in *Babot et al., 2014*). It is currently unknown whether plant CI displays an active-to-deactive transition.

The alternative NDs irreversibly operate in the forward directly over the large range of [NAD$^+$]/[NADH] in which CI operates in reverse. As discussed above, this would also be the case for a CI*

that pumped protons at $2H^+$:$2e^-$. The simultaneous activity of alternative NDs and CI would continuously push the [NAD$^+$]/[NADH] ratio towards the CI RET regime, due to the irreversible oxidation of NADH and reduction of CoQ by the NDs. Thus, the potential existence of a $2H^+$:$2e^-$-pumping CI* does not generate additional bioenergetic problems beyond those already created by the existence of the alternative NDs (which do not pump any protons at all). The plant cell must already have regulatory mechanisms to deal with the threat of RET by CI imposed by the NDs. The degree to which these alternative NDs are employed and regulated *in vivo* remains poorly understood (*O'Leary et al., 2019*). We predict that some type of rectification operates on plant CI as a mechanism to prevent ROS production under any conditions that favors RET by CI.

This analysis also proposes a possible answer to why our preparations of etiolated *V. radiata* contain such a significant amount of CI*, compared to the previously reported lower abundance of CI* in non-etiolated tissues (*Ligas et al., 2019*; *Senkler et al., 2017*). To the best of our knowledge, the mitochondrial [NAD$^+$]/[NADH] ratio of etiolated hypocotyls has not been investigated. However, given the lack of input of reducing equivalents by the C$_2$ cycle via GDH in the dark (an otherwise high-flux pathway), it is conceivable that the [NAD$^+$]/[NADH] ratio in etiolated hypocotyls is higher than in photosynthesizing cells. A high [NAD$^+$]/[NADH] ratio may favor the use of CI* over CI in order to ensure maintenance of the proton motive force ($\Delta$p), at the expense of thermodynamic efficiency. It is conceivable that, as hypocotyls develop under etiolating conditions and their only source of energy (i.e. the seed oils) diminishes, the ratio of CI* to CI present in the mitochondrial membranes may be dynamically regulated to increase CI* levels.

Although CI*'s proton-pumping ratio remains to be characterized, the theoretical analysis above suggests that that a $2H^+$:$2e^-$-pumping entity may be beneficial for plants' bioenergetic flexibility if a rectification mechanism for CI exists in plants. Further studies are needed to test these hypotheses.

