## [Decision Letter]

**Acceptance summary:**

This structural work represents a significant advance in our understanding of the mechanism and regulation of mitochondrial Complex I, particularly in plants, which have been relatively little studied. Further, the system that was developed will serves the field more generally by providing a useful platform for exploring a broad range of open questions about the function, regulation, assembly and Complex I in plants.

**Decision letter after peer review:**

Thank you for submitting your article "Atomic Structure of a Mitochondrial Complex I Intermediate from Vascular Plants" for consideration by *eLife*. Your article has been reviewed by three peer reviewers, one of whom is a member of our Board of Reviewing Editors, and the evaluation has been overseen by Cynthia Wolberger as the Senior Editor. The following individuals involved in review of your submission have agreed to reveal their identity: Toshiharu Shikanai (Reviewer #2); Nicholas Fisher (Reviewer #3).

The reviewers have discussed the reviews with one another and the Reviewing Editor has drafted this decision to help you prepare a revised submission. Your manuscript is of interest, but as described below that additional experiments are required before it is published. We would like to draw your attention to changes in our revision policy that we have made in response to COVID-19 (https://elifesciences.org/articles/57162). First, because many researchers have temporarily lost access to the labs, we will give authors as much time as they need to submit revised manuscripts. We are also offering, if you choose, to post the manuscript to bioRxiv (if it is not already there) along with this decision letter and a formal designation that the manuscript is “in revision at *eLife*”. Please let us know if you would like to pursue this option. (If your work is more suitable for medRxiv, you will need to post the preprint yourself, as the mechanisms for us to do so are still in development.)

Summary:

The structural work is of high quality and has interesting, possibly strong, implications for the mechanism of Complex I (CI) and its roles in diverse systems. The work focuses on a subcomplex, CI*, that is thought to be an assembly intermediate, and is missing the distal subcomplex of the proton-pumping component of the complex. An interesting hypothesis is presented in which CI* is functional, but has lower capacity to pump protons, allowing the mitochondrion to modulate energy coupling.

Essential revisions:

All reviewers had concerns about the relevance of the CI* subcomplex, and felt that it is critical to assess, at least to some extent, whether it is functional. Indeed, the title of the paper and part of the discussion assumes that CI* is an assembly intermediate, while much of the Discussion centers around the possibility that CI* is functional, but with a lower H^+^ stoichiometry. The manuscript does not provide any support for either of these roles.

While the reviewers recognize the difficulty in obtaining new data in the current COVID-19 crisis, knowing if the subcomplex is a functional component, a true assembly intermediate, both or an artifact of isolation is essential for the interpretation of much of the results.

Reviewers #2 and 3 provide straightforward approaches to test for these roles. First, it was thought important to address the question of whether the CI* subcomplex is an artifact of isolation, and if more intact complexes were isolated in other tissues. Second, it is essential to determine if the CI* complex has enzymatic activity. Although not considered to be essential, the paper would have greater impact if it was possible to demonstrate a change in H^+^/e^-^ pumping stoichiometry. The title and Discussion should be modified to in response to these additional data.

Reviewer #1:

This manuscript describes a cryo-EM structure of a portion of plant mitochondrial Complex I (1). This subcomplex, termed CI*, is thought to represent an intermediate in assembly.

The work shows key details about some of the structural distinctions between plant and non-plant CI, including the additional subunit related to carbonic anhydrase (CA). The analysis of the structure leads to some interesting discussion, but it lacks firm hypothesis testing that would allow for stronger conclusions.

1) "Whereas the exact γCA protein combinations are likely tissue- and development-stage-dependent (Cordoba et al., 2019), the role of the γCA domain in plant CI's function is unknown (Martin et al., 2009)."

How does this relate to the CA thought to be involved in the CCM in cyanobacteria?

2) How do we know if this is really the CI* intermediate and not some sort of breakdown product?

3) I am confused by this:

“At each active site, two histidine residues are provided by one subunit and the third is provided by the adjacent subunit. However, in the plant CI γCA heterotrimer, the γCA2L subunit is lacking two of the three essential histidine residues (Ala-147 and Arg-152 in *V. radiata*) that would be necessary to form active sites at the interfaces with the CA1 and CA2 subunits. This renders two of the possible three catalytic sites non-functional (Figure 3A, Figure 3—figure supplement 2) …Our subunit assignments are consistent with this nonfunctional residue on CA1 being at the interface with CA2L, which would already be inactive due to the lack of histidine on CA2L at position."

It is unclear is the structure provides independent support that there is one of three active sites, or is their assignment based on this assumption?

In other words, there are hypotheses that the proteins represent one-of-three active CA or are completely inactive for CA activity. To what extent can the new structure actually test these possibilities? The test states that the Zn^2+^ was observed only one site. How strong is this evidence and how strongly does this support the one-in-three CA model? In this regard is it far more important what is inconsistent with the structure than what can be made to be consistent with it. In other words, if the structure can eliminate all but one of these hypotheses, it would be a big advance.

4) "Therefore, carbonic anhydrase activity of the plant γCA domain on CI must be confirmed experimentally." Why not test for activity in the amazing new prep?

5) "The lack of such a cap on *V. radiata* NU2M in CI* suggests that, although the Lys399 of NU2M is mostly surrounded by protein, the core hydrophilic axis may be in contact with lipid." Alternatively, some subunits may be missing or difficult to resolve in the prep.

6) "This leaves open the possibility that the γCA domain does not interconvert CO_2_ and bicarbonate but acts only as a sensor of CO_2_ or bicarbonate concentration. In this scenario, conformational changes induced by bicarbonate binding could propagate into the membrane arm of CI and potentially regulate the catalytic turnover of CI. Such sensory and regulatory roles have also been proposed for other CI accessory subunits in other organisms."

Is there any evidence from the structure in support of are inconsistent with the proposed conformation propagation mechanism for CO_2_ sensing? Perhaps I am missing something, but the CA subunit seems to be rather loosely tethered, and it is difficult to see from the images provided how small conformational changes at the metal site would strongly affect the structure.

Also, given that this is a major point of the discussion, elaborate on why such signaling is interesting and what does the structure say about it. In particular, see points above on whether the structure confirms or not the proposed lack of catalytic activity.

7) The remaining differences between this and other structures are framed as important for assembly. Yet, how is it known that this is a real assembly intermediate (see point 2)?

8) I am totally not convinced that there is any evidence from the structure one way or the other for a variable H^+^/e^-^ transmission. Please convince me.

9) "In heterotrophs that such as metazoans that obtain all their reduction equivalents from the breakdown of sugars, fats and proteins and that depend almost entirely on oxidative phosphorylation for energy production, there would be no advantage to being able to adjust CI's H^+^-pumping-to-electron-transfer ratio." This statement seems to be contradicted by the text that follows, which argues that adjusting H^+^/e^-^ stoichiometry might be important for maintaining PMF (though I am not sure I agree with those).

10) Regarding the paragraph from "In autotrophs such as plants, which generate their energy through a combination of respiration and photosynthesis, the bioenergetic processes necessarily have to be more dynamic to balance production of energy (ATP) and of reducing equivalents (NADH) throughout the cell and the organism."

This is highly speculative, which in itself is OK, but I'm not sure what, if anything in the structure, supports this view.

11) The authors are correct that the energy balance of the cell will be strongly impacted by activating photosynthesis. However, there are several factors that need to be considered: 1) It is known for some time that respiration is down-regulated during photosynthesis, so that little if any ATP is produced by the mitochondria; 2) The critical factor is often the balance of energy in the forms of ATP/ADP+Pi and NAD(P)H/NADP+ rather than the total energy availability. In this case, a variable proton pumping stoichiometry might be useful, but; 3) There are already alternative mechanisms of shunting electrons to O_2_ without production of ATP, e.g. though the alternative oxidases or Water-water cycles.

12) "We posit that, in addition to bypassing H^+^-pumping altogether through the alternative complexes, it is conceivable that plants may find additional advantages in regulating the H^+^-pumping-to-electron-transfer ratio in the "canonical" electron transport chain. If the NADH/NAD+ ratio dropped in photosynthetic cells due to decreased light availability, this would lead to a decrease in free energy available to CI for H^+^ pumping.

13) "Full-length CI must pump 4 protons per 2 NADH electrons, it cannot operate in a graduated manner. Consequently, at a decreased NADH/NAD+ ratio, with lower free energy available, full-length CI would initially not be able to pump any protons at all. This would result in a reduction in the mitochondrial membrane potential, decreasing the proton-motive force and lowering ATP generation.

First, I think this point, if a full-length CI can pump lower Hstoichiometry of H^+^/e^-^ is debatable (and being debated), so one cannot conclude that it cannot.

Second, the logic connecting the two sentences above seems to be incorrect. The smaller the coupling stoichiometry, the higher (not lower) the PMF that can be maintained at static head for a given NADH/NAD+ ratio.

CI is a reversible proton pumping-NADH:UQ oxidoreductase, so it will reach quasi-equilibrium (static head) when the free energy in the reaction

NADH + UQ + nH(n)+ ⇓ ◊ NAD+ UQH2 + nH^+^(p)

equals that in PMF.

The text then states: "however, if the plant ETC could find a way to keep pumping protons even at a decreased NADH/NAD+ ratio…" This would seem to be self-contradictory. One possibility is that the authors intend to include the total proton translocation of the system, and not just "pumping". In this case, PMF includes contributions from the chemical protons that are taken up during UQ reduction, which are likely to be deposited on the p-side of the membrane during UQH2 oxidation etc, e.g. at the Q-cycle.

The text continues, "however, if the plant ETC could find a way to keep pumping protons even at a decreased NADH/NAD+ ratio, this would allow for better maintenance of the proton motive force." The authors are apparently conflating near equilibrium and steady-state systems. With a smaller coupling stoichiometry, the number of protons pumped per NADH oxidized decreases, but the extent of PMF that can be generated at near equilibrium conditions increases. In a situation where the PMF is far out of equilibrium from the free energy in oxidation of NADH by UQ, however, the PMF is determined by the competition between proton pumping by CI and that by the ATP synthase. If this is the case, and CI rates are limiting, one might expect to see a decrease in PMF as pumping stoichiometry decreases.

14) "Although the enzymatic activity of CI* and its H^+^-pumping-to electron-transfer ratio still remain to be confirmed, we venture that, by potentially pumping protons at a 2H^+^ :2e^-^ ratio, CI* may play novel roles in the regulation of electron flow and energy conservation, adding flexibility to the mitochondrial electron transport chain of plants."

First, I object to the term "novel" in this context. How can one claim novelty when one does not know how all the others work?

15) "These observations also suggest that, throughout eukaryotic evolution, different bioenergetic strategies have demanded a trade-off between respiratory efficiency and dynamism." What observations are being referred to here?

It seems to be inferred, but never clearly stated, that CI* is a true, active form of CI, but with half of the proton pumping capacity.

The only place this is really clearly stated are in statements like "this suggests that there are significant steady-state amounts of this assembly intermediate in *V. radiata* mitochondria under these conditions and that CI* may be playing an independent physiological function."

Yet, the text also seems to imply that this form is an assembly intermediate. In any case, if this is an active intermediate, they should demonstrate this with at least a rudimentary activity assay.

16) The text starts to speculate about "dynamism", which is an interesting topic worthy of a review/hypothesis paper, but the data does not really provide very strong arguments in favor of such flexibility. If the authors want to make the case that their structure does indicate such flexibility, they will need to bolster their analysis.

On the other hand, there are some interesting observations in the data, especially the CA subunit, that could provide the bases of a more powerful discussion.

Reviewer #2:

An assembly intermediate of the mitochondrial complex I (CI*) was isolated from etiolated seedlings of mung bean and its structure was determined at 3.9 A resolution. The structure revealed some diversity from the complexes of bacteria, yeast and mammals including the presence of γCA subunits. Based on the structure, they propose that the CI* may be functional and provides the flexibility on respiratory electron transport.

This is the first report of the structure of plant complex I (an assembly intermediate) and provides important scientific message. The quality of science is high, and text is written clearly in general. I have several opinions to improve the manuscript.

Specific comments

1) The authors did not provide any evidence on the CI* to be functional. The idea of the regulation of the H^+^/e^-^ ratio is attractive but the discussion sounds rather speculative and redundant (the last four paragraphs).

2) Plant mitochondria function in consuming excessive reducing power generated in chloroplasts and the proposed idea is interesting. But the CI* complexes were isolated from etiolated seedlings. Is it possible to detect the CI* complexes in green tissues?

3) They did not show the activity of the CI* complex. Is there any genetic evidence supporting that the complex I is functional in the absence of PD?

4) I am unsure why they focused on the assembly intermediate before clarifying the structure of the fully assembled complex I.

5) P5, L18. NUM2?

Reviewer #3:

The authors present a cryoEM structure at a nominal 3.9 Å resolution for an intermediary form of mitochondrial respiratory Complex I isolated from etiolated mung bean tissue. This is an exciting and timely manuscript and represents the first report of a (pseudo)atomic structure of Complex I (CI) from plants.

I am, in general, enthusiastic about this manuscript and consider it within the remit for publication within *eLife*. It is well written, and the description of the structure especially seems very sound. I do, however, have questions for the authors concerning the methodology, and in particular, the discussion.

i) The enzyme, as isolated from etiolated mung bean by detergent treatment, is lacking two major (proton-translocating) subunits from the membrane domain (the “PD” domain) and is described throughout the manuscript as CI*, as an intermediate, assembly form of Complex I. This is, in principle, acceptable, as this CI* intermediate has been documented by others elsewhere. However, it does raise the question whether this is an artefact of isolation (i.e. JBC 278: 43114 , 2003). Did the authors observe any intact CI in their preparation, or evidence for a solubilised NuoL/M subcomplex? It also remains to be seen if CI* is a peculiarity to etiolated tissue, and perhaps the authors can expand on this.

ii) The enzymatic activity of the CI* preparation was not determined. The authors suggest, that it functions as a 2H^+^/2e pump, by analogy with mutant forms of the fungal enzyme. This in itself is a reasonable suggestion, as this stoichiometry has also been observed with similar mutants of *E. coli* Complex I (Biochemistry 50: 3386, 2011), however, it is a pity that this was not investigated here as a discussion of the bioenergetics of CI* form a central part of the manuscript.

iii) The authors write that "Moreover, in photosynthetic tissue, conditions of intense light may lead to an over-production of NADH". I think this statement needs qualification for the benefit of non-expert readers. Obviously, the primary reductant produced by photosynthetic linear electron flow is NADPH and plants have many mechanisms to deal with sudden increases in light intensity (induction of NPQ, photosynthetic control at the level of cytochrome b6f etc.). Now, it is true that “excess” reducing equivalents are transferred from the chloroplast via the malate shunt, but this is a relatively low flux pathway, and so I assume that the authors are referring to NADH produced by the photorespiratory pathway via glycine decarboxylase activity. It is reasonable to assume that this may cause relatively large fluctuations in intramitochondrial [NADH]. (This also, obviously, will not be a factor in non-photosynthetic/etiolated tissue).

Related to point ii) above, the authors write "If the NADH/NAD+ ratio dropped in photosynthetic cells due to decreased light availability, this would lead to a decrease in free energy available to CI for H^+^ pumping. Full-length CI must pump 4 protons per 2 NADH electrons, it cannot operate in a graduated manner. Consequently, at a decreased NADH/NAD+ ratio, with lower free energy available, full-length CI would initially not be able to pump any protons at all. This would result in a reduction in the mitochondrial membrane potential, decreasing the proton-motive force and lowering ATP generation."

The thinking seems muddled here. Firstly, Complex I can be assumed to be operating in equilibrium (or near equilibrium) with the PMF, and so a “variable” H^+^/e pumping stoichiometry afforded by CI*/CI provides no benefit. The authors can demonstrate this for themselves through elementary thermodynamic principles if they seek reassurance (i.e. the equilibrium relationships (H^+^/2e)*PMF = 2ΔE_h_ and also K = 10^((ΔE_h_-(PMF*(H^+^/2e))/30) etc) – in fact, I would encourage this as a supplementary figure. I doubt that there will be circumstances under which a supply of reducing equivalents to the plant mitochondrial respiratory chain will become limiting, particularly given the menagerie of electron donors that supply it. I can think of one instance in which 2H^+^/2e translocating CI* would provide a benefit over a 4H^+^/2e counterpart; it would be effectively irreversible at high (>200 mV) PMF, which may limit ROS production due to reverse ET (clearly the non-protonmotive NDA enzymes can be considered irreversible under all circumstances).

iv) The suggestion that the γCA subunit may be acting as a bicarbonate sensor by conformational change transmitted to the membrane domain seems very speculative.

v) Typo – Figure 1—figure supplement 3 – “accesory” in many places.

---

## [Author Response]

Reviewer #1:[…]1) "Whereas the exact γCA protein combinations are likely tissue- and development-stage-dependent, the role of the γCA domain in plant CI's function is unknown."How does this relate to the CA thought to be involved in the CCM in cyanobacteria?

The carbonic anhydrase family is a highly diverse family of enzymes, composed of seven genetically distinct classes. While the seven classes possess carbonic anhydrase/CO_2_ hydratase activity, the structure and domain composition of the seven classes is diverse, and the sequence similarity across families is low. In α- and γ-CA classes, the metal ion in the active site is coordinated by three histidine residues. In contrast, the metal is coordinated by one histidine and two cysteine residues in the β-CA class. Moreover, α-CAs are active as monomers or dimers and β-CAs (which have a distinct structure compared to α-CAs) are active as tetramers. Finally, γ-CAs have a characteristic left-handed parallel β-sheet fold and are only active as trimers with active sites typically between subunits.

Genomic and phylogenetic studies have shown that bacteria may encode for α, β or γ CAs. In contrast, cyanobacteria have only been proven to encode for α- or β-CA in their various CO_2_-concentrating mechanisms (CCM). Moreover, searches of DNA databases of vascular plants and algae have not found clear homologues of cyanobacterial CCM genes. It is therefore believed that CCM genes, including those for cyanobacterial CAs, are almost exclusively restricted to cyanobacteria (and some proteobacteria). Nevertheless, cyanobacterial CA proteins involved in CCM do belong to the large CA protein family.

Therefore, the association seen between CA proteins and CI in cyanobacteria and plants is a case of convergent evolution, the physiology of which is incompletely understood. Given the recency of the first structures of cyanobacterial CI (Schuller et al., 2020) and plant CI (presented here), it still remains unclear whether the association of CI and CA in the two domains of life was driven to fulfil the same physiological function.

2) How do we know if this is really the CI* intermediate and not some sort of breakdown product?

The reviewer brings up an important point, which we should have emphasized in our manuscript. We present the arguments that support that CI* is an assembly intermediate and not a degradation product below. We have included these points in a new section in the Discussion of our revised manuscript (Discussion – Protein sample) and have added a new supplementary figure showing details of CI*’s purification and enzymatic activity (Figure 1—figure supplement 2).

We believe our sample is the CI* and not a degradation product for the following reasons:

– The 800-kDa CI subcomplex missing the peripheral-pumps domain (i.e. CI*) is a known subcomplex of plant CI, as determined by genetic and complexome profiling experiments of CI’s assembly pathway in *Arabidopsis thaliana* (Meyer et al., 2011, Schertk et al., 2012, Schimmeyer et al., 2016, Senkler et al., 2017, Ligas, et al., 2019). This intermediate is seen in non-etiolated seedlings and mature leaves in *A. thaliana* and *N. sylvestris* (Pineau et al., 2008). Moreover, the *A. thaliana* and *N. sylvestris* CI* showed NADH-dehydrogenase activity by the same in-gel activity assay we use in our preparation (Meyer et al., 2011, Pineau et al., 2008).

– The membrane-extraction conditions used in our prep (1% w:v digitonin, 4:1 g:g detergent:protein) are very gentle conditions for membrane protein extraction. These gentle conditions were chosen after optimization of different percentages, amounts and types of detergent to preserve protein:protein interactions in protein complexes.

– Immediately after extraction, we trap the detergent-extracted complexes in amphipatic polymers, which stabilize the complexes and protect them from degradation/dissociation (Breibeck and Rompel, 2019).

– These conditions are sufficiently gentle that they do not fully dissociate a large portion of lipid membrane that co-purifies with CI* and can be seen at low contour in our reconstructions (Figure 1—figure supplement 4E).

– Using the same detergent, digitonin, an *A. thaliana* complexome profiling study (Senkler, 2017) not only obtained full-length CI and CI*, but also full-length CI in a higher-order assembly with complex III, i.e. the “supercomplex” SC I+III_2_. The protein:protein interactions between complexes in a supercomplex more labile than intra-complex protein:protein interactions. Given that the more fragile CI:CIII_2_ interactions are maintained in 5% digitonin (Senkler et al., 2017), this argues that the presence of CI* – both in the Senkler and in our study – is not due to digitonin-induced dissociation of the distal domain, but rather that it is the true assembly intermediate.

– Published controlled-degradation experiments of plant CI in the presence of increasingly harsh detergents (increasingly strong disruption of protein:protein interactions within the complex) have shown that, similar to mammalian CI, plant CI detergent-induced dissociation occurs by dissociation of the full peripheral arm from the full matrix arm (Klodmann et al., 2010). In other words, CI detergent dissociation occurs at the “elbow” between CI’s arms and not by dislodging the distal pumps from the matrix-arm-associated-proximal pumps, which would need to occur if CI* were a degradation product.

– We have reproducibly obtained the CI* fraction, which retains in-gel and spectroscopic NADH-oxidase activity and chromatographic peak even after several days of manipulation. These data have been included in the new Figure 1—figure supplement 2.

3) I am confused by this:“At each active site, two histidine residues are provided by one subunit and the third is provided by the adjacent subunit. However, in the plant CI γCA heterotrimer, the γCA2L subunit is lacking two of the three essential histidine residues (Ala-147 and Arg-152 in V. radiata) that would be necessary to form active sites at the interfaces with the CA1 and CA2 subunits. This renders two of the possible three catalytic sites non-functional (Figure 3A, Figure 3—figure supplement 2) …Our subunit assignments are consistent with this nonfunctional residue on CA1 being at the interface with CA2L, which would already be inactive due to the lack of histidine on CA2L at position."It is unclear is the structure provides independent support that there are one of three active sites, or is their assignment based on this assumption?In other words, there are hypotheses that the proteins represent one-of-three active CA or are completely inactive for CA activity. To what extent can the new structure actually test these possibilities? The test states that the Zn^2+^ was observed only one site. How strong is this evidence and how strongly does this support the one-in-three CA model? In this regard is it far more important what is inconsistent with the structure than what can be made to be consistent with it. In other words, if the structure can eliminate all but one of these hypotheses, it would be a big advance.

We thank the reviewer for pointing out these confusing statements. We have re-worded the revised document to clarify this issue.

The structure provides independent support that only one of the putative active sites could be active, as there is only one site where all three Zn-coordinating residues are present. The EM map shows clear density for the three histidine sidechains and the Zn^2+^ ion at the active site 1. Likewise, the density for the sidechains in active sites 2 and 3 is also clear. It is unambiguous that no histidine sidechain density is visible at active sites 2 and 3. Chemically, no Zn^2+^ ion could be coordinated by the residues in active sites 2 and 3. Consequently, no density for the Zn^2+^ ion is seen in those locations. This structural data provides independent confirmation of our subunit assignment from our sequence alignments. The structure eliminates all but one possible subunit arrangement to fit in the density seen and shows that only one of the three potential active sites allows for Zn^2+^ coordination and could potentially be actually enzymatically active.

4) "Therefore, carbonic anhydrase activity of the plant γCA domain on CI must be confirmed experimentally." Why not test for activity in the amazing new prep?

Once we are able to return to the lab, we are indeed planning on testing the potential carbonic anhydrase activity of CI* and CI, as well as the potential interplay between CO_2_ sensing and CI turnover. However, this is beyond the scope of this current manuscript.

5) "The lack of such a cap on V. radiata NU2M in CI* suggests that, although the Lys399 of NU2M is mostly surrounded by protein, the core hydrophilic axis may be in contact with lipid." Alternatively, some subunits may be missing or difficult to resolve in the prep.

Although it is theoretically possible that a subunit at the end of the proximal domain may have been lost, given our gentle extraction conditions and our exchange into amphipol, this is unlikely, as discussed in response to reviewer’s point 2. We agree with the reviewer that it is surprising to observe a lack of capping subunits on the end of the membrane arm, which is why we qualify the statement carefully with “suggests that” and “may be.” Nonetheless, the EM density does not show the presence of any subunits in this location – not even weak density that could be interpreted as a hard-to-resolve subunit – and we must report what we observe.

6) "This leaves open the possibility that the γCA domain does not interconvert CO_2_ and bicarbonate but acts only as a sensor of CO_2_ or bicarbonate concentration. In this scenario, conformational changes induced by bicarbonate binding could propagate into the membrane arm of CI and potentially regulate the catalytic turnover of CI. Such sensory and regulatory roles have also been proposed for other CI accessory subunits in other organisms."Is there any evidence from the structure in support of are inconsistent with the proposed conformation propagation mechanism for CO_2_ sensing? Perhaps I am missing something, but the CA subunit seems to be rather loosely tethered, and it is difficult to see from the images provided how small conformational changes at the metal site would strongly affect the structure.Also, given that this is a major point of the discussion, elaborate on why such signaling is interesting and what does the structure say about it. In particular, see points above on whether the structure confirms or not the proposed lack of catalytic activity.

Since the initial biochemical experiments on plant’s CA domain (Sunderhaus et al., 2006), it has been recognized that the CA domain is tightly linked to the membrane arm. Our structure shows that the interface between the CA domain and the membrane arm is extensive, with protein:protein interactions of the CA domain with several of the membrane-arm subunits (NU2M, NDUC2, P2, NDUX1). We were able to quantify the size and strength of the CA-membrane arm interface using the standard PDBePISA protein interface tool. We estimate that the interface between the CA domain and the membrane arm is ~3740 A^2^, with a solvation energy gain of ~210 kcal/mol. This gain in free energy is almost twice as large as the gain from the association of the CA domain itself, demonstrating that it is a tight interaction.

We have added these points to our description of the CA domain in the main text and have included a new supplementary table with the details of the interface size and strength (Table 5).

Given this large surface and the number of protein:protein interactions, it is feasible that a conformational change in the CA domain could propagate to the N or Q modules to affect NADH-ubiquinone oxidoreduction. However, our structure does not provide evidence for the catalytic activity of the CA domain (a long-standing hypothesis in the field) or for any potential influence of CA conformational changes on CI’s NADH-ubiquinone oxidoreduction. We have therefore removed the speculation about the potential carbonate sensing from our discussion.

7) The remaining differences between this and other structures are framed as important for assembly. Yet, how is it known that this is a real assembly intermediate (see point 2)?

CI* is a known assembly intermediate of plant CI, as discussed in response to point 2).

8) I am totally not convinced that there is any evidence from the structure one way or the other for a variable H^+^/e^-^ transmission. Please convince me.

The nature of CI* is that it lacks two proton pumps (those forming the distal domain) compared to full-length CI, which contains four proton pumps and is known to pump protons at a 4:2 proton:electron ratio. Our structure shows the molecular arrangement of CI*, which contains all of the subunits and cofactors needed for the electron transfer reactions between NADH and CoQ but lacks two out of four proton-pumps. It is correct that our structure does *not* show that CI* pumps protons. However, the structure does strongly imply that if CI* were capable of pumping protons, it would be at a lower proton-to-electron ratio than full-length CI due to the lack of two proton pumping domains. This is a hypothesis at this stage, and its testing requires functional experiments. Nonetheless, genetic manipulations of CI in *Y. lipolytica* and *E. coli* have resulted in equivalent CI subcomplexes lacking the two distal H^+^-pumps. These CI subcomplexes are capable of NADH-CoQ oxidoreduction and proton-pumping at a 2:2 proton:electron ratio, rather than the 4:2 proton:electron ratio seen for full length CI (Drose et al., 2011, Steimle et al., 2011). Given these analogous studies, we conclude that given our structure it is reasonable to hypothesize that CI* could also be a functional subcomplex that pumps protons at a 2:2 ratio.

Moreover, in this revised manuscript, we add a new figure with biochemical data showing that CI* is capable of NADH-ubiquinone oxidoreduction (Figure 1—figure supplement 2). We have also re-written our discussion on CI*s potential lower proton:electron ratio to shorten the section and emphasize its speculative nature. As suggested by reviewer 3, we have moved the bioenergetic discussion into an Appendix, where we provide a more detailed and quantitative explanation of our rational for the possible role of CI* in plants, to address the reviewers’ bioenergetic concerns.

We intend to more thoroughly characterize the function of CI* and carefully quantify its proton:eletron ratio using a reconstituted system. However, given the competitive situation surrounding this manuscript and the pandemic-related limitations on wet-lab research, we believe it is better to develop this evidence in a follow-up manuscript focusing on a deep functional and enzymatic characterization of CI*.

9) "In heterotrophs that such as metazoans that obtain all their reduction equivalents from the breakdown of sugars, fats and proteins and that depend almost entirely on oxidative phosphorylation for energy production, there would be no advantage to being able to adjust CI's H^+^-pumping-to-electron-transfer ratio." This statement seems to be contradicted by the text that follows, which argues that adjusting H^+^/e^-^ stoichiometry might be important for maintaining PMF (though I am not sure I agree with those).

The statement above presented the situation in heterotrophs in contrast with that in autotrophs. Nonetheless, for clarity and to focus more on our structural findings, we have removed these paragraphs from this section of the Discussion.

10) Regarding the paragraph from "In autotrophs such as plants, which generate their energy through a combination of respiration and photosynthesis, the bioenergetic processes necessarily have to be more dynamic to balance production of energy (ATP) and of reducing equivalents (NADH) throughout the cell and the organism. "This is highly speculative, which in itself is OK, but I'm not sure what, if anything in the structure, supports this view.

We have re-written our Discussion to better align with the results from our structure and to remove unnecessary speculation. We have lengthened and made a separate section on the CA domain and shortened the speculative section on CI*’s potential roles. Moreover, we have included an Appendix to further explain our bioenergetic arguments for the possible roles of CI*.

11) The authors are correct that the energy balance of the cell will be strongly impacted by activating photosynthesis. However, there are several factors that need to be considered: 1) It is known for some time that respiration is down-regulated during photosynthesis, so that little if any ATP is produced by the mitochondria; 2) The critical factor is often the balance of energy in the forms of ATP/ADP+Pi and NAD(P)H/NADP+ rather than the total energy availability. In this case, a variable proton pumping stoichiometry might be useful, but; 3) There are already alternative mechanisms of shunting electrons to O_2_ without production of ATP, e.g. though the alternative oxidases or Water-water cycles.

We agree with the reviewer, although our understanding is that point 1 remains controversial, as recent experimental results and metabolic simulations suggest an important role for mitochondria in the supply of ATP to the cytosol under conditions in which the ATP consumption in photosynthetic CO_2_ fixation is sufficiently high (see references below). We have clarified these points in the Discussion and clarified the discussion in an Appendix.

12) "We posit that, in addition to bypassing H^+^-pumping altogether through the alternative complexes, it is conceivable that plants may find additional advantages in regulating the H^+^-pumping-to-electron-transfer ratio in the "canonical" electron transport chain. If the NADH/NAD+ ratio dropped in photosynthetic cells due to decreased light availability, this would lead to a decrease in free energy available to CI for H^+^ pumping.13) "Full-length CI must pump 4 protons per 2 NADH electrons, it cannot operate in a graduated manner. Consequently, at a decreased NADH/NAD+ ratio, with lower free energy available, full-length CI would initially not be able to pump any protons at all. This would result in a reduction in the mitochondrial membrane potential, decreasing the proton-motive force and lowering ATP generation.First, I think this point, if a full-length CI can pump lower Hstoichiometry of H^+^/e^-^ is debatable (and being debated), so one cannot conclude that it cannot.

We have removed these paragraphs and clarified our bioenergetic arguments in an Appendix.

Second, the logic connecting the two sentences above seems to be incorrect. The smaller the coupling stoichiometry, the higher (not lower) the PMF that can be maintained at static head for a given NADH/NAD+ ratio.CI is a reversible proton pumping-NADH:UQ oxidoreductase, so it will reach quasi-equilibrium (static head) when the free energy in the reactionNADH + UQ + nH(n)+ ⇓ ◊ NAD+ UQH2 + nH^+^(p)equals that in PMF.The text then states: "however, if the plant ETC could find a way to keep pumping protons even at a decreased NADH/NAD+ ratio…" This would seem to be self-contradictory. One possibility is that the authors intend to include the total proton translocation of the system, and not just "pumping". In this case, PMF includes contributions from the chemical protons that are taken up during UQ reduction, which are likely to be deposited on the p-side of the membrane during UQH2 oxidation etc, e.g. at the Q-cycle.The text continues, "however, if the plant ETC could find a way to keep pumping protons even at a decreased NADH/NAD+ ratio, this would allow for better maintenance of the proton motive force." The authors are apparently conflating near equilibrium and steady-state systems. With a smaller coupling stoichiometry, the number of protons pumped per NADH oxidized decreases, but the extent of PMF that can be generated at near equilibrium conditions increases. In a situation where the PMF is far out of equilibrium from the free energy in oxidation of NADH by UQ, however, the PMF is determined by the competition between proton pumping by CI and that by the ATP synthase. If this is the case, and CI rates are limiting, one might expect to see a decrease in PMF as pumping stoichiometry decreases.

We agree with the reviewer that this discussion was unclear. In this revised manuscript, we have made this discussion more succinct in the main text. In order to better express our arguments, we have included a more quantitative discussion in an Appendix.

We fully appreciate the distinctions pointed out by the reviewer between near-equilibrium and steady-state limitations. We address this in more detail in the Appendix to better support our claims about the possible roles of CI*.

The steady-state limit of the proton motive force (PMF) was determined in etiolated *V. radiata* mitochondria to be 200 mV (Moore and Bonner in 1981). Using reasonable assumptions (see Appendix for details), it is possible to determine that pumping 4 protons against a 200 mV PMF would require an [NAD+]/[NADH] ratio of 0.3 or less (i.e. the pool would have to be >70% reduced). Conversely, CI* pumping 2 protons would work in the forward direction against a 200 mV PMF up to a [NAD+]/[NADH] of >105. As the reviewer states, if CI is rate-limiting under these conditions, a decrease in pumping stoichiometry may result in a decrease in PMF, as ATP synthase outpaces the supply of protons. Nonetheless, the presence of a functional CI* under conditions of high [NAD+]/[NADH], even if rate-limiting, would still maintain the PMF to a greater extent than if all proton-pumping by CI ceased due to the loss of free energy of NADH oxidation. The degree to which this would be the case requires a steady-state model of the plant electron transport chain and would need good estimates of stoichiometries of the plant mitochondrial electron transport complexes (including CI*) and ATP synthase which currently (to the best of our knowledge) are not available and go beyond the scope of this paper.

Of course, 200 mV is an upper limit of steady-state proton pumping and likely the PMF in the plant cell is somewhat lower. Nonetheless, under any conditions in which full CI is operating near equilibrium, a two-proton-pumping CI* would be irreversible. This is more so the case for the non-proton pumping alternate NADH dehydrogenases (NDs). The point we failed to make clearly in the original manuscript was that if CI is operating near equilibrium and the [NAD+]/[NADH] ratio increases due to metabolic fluctuations (perhaps even due to changes in light, due to the input of the glycine dehydrogenase complex into mitochondrial NADH levels via the C2 photorespiration cycle), the PMF will decrease. The presence of CI* could be used to maintain the PMF at or closer to steady-state levels (with a 50% loss in thermodynamic efficiency) under a large range of [NAD+]/[NADH] ratios that favor reverse electron transport by full CI. Of course, many questions still remain, and this could lead to other issues of reactive oxygen species by CI via reverse electron transport. However, this problem already exists in plant mitochondria owing to the activity of NDs, which have an even stronger thermodynamic drive to push the energetics towards CI reverse electron transport. Many unknowns remain with respect to the co-expression of CI, the NDs – and potentially CI* – in mitochondrial membranes.

Moreover, the levels of CI* in our etiolated *V. radiata* sprouts appear higher than in reports from non-etiolated samples. We suspect this may be due to the seedlings’ major energy source (seed oils) being consumed, and cellular NADH levels dropping without energetic input from photosynthesis. This may favor a push towards less efficient but forward proton-pumping through the use of CI*. This will need to be tested experimentally and is beyond the scope of this manuscript.

14) "Although the enzymatic activity of CI* and its H^+^-pumping-to electron-transfer ratio still remain to be confirmed, we venture that, by potentially pumping protons at a 2H^+^ :2e^-^ ratio, CI* may play novel roles in the regulation of electron flow and energy conservation, adding flexibility to the mitochondrial electron transport chain of plants."First, I object to the term "novel" in this context. How can one claim novelty when one does not know how all the others work?

We take the reviewer’s point. We removed this sentence.

15) "These observations also suggest that, throughout eukaryotic evolution, different bioenergetic strategies have demanded a trade-off between respiratory efficiency and dynamism." What observations are being referred to here?

In order to improve the clarity of the manuscript and reduce speculation, we have removed this discussion from the manuscript.

It seems to be inferred, but never clearly stated, that CI* is a true, active form of CI, but with half of the proton pumping capacity.The only place this is really clearly stated are in statements like "this suggests that there are significant steady-state amounts of this assembly intermediate in V. radiata mitochondria under these conditions and that CI* may be playing an independent physiological function."Yet, the text also seems to imply that this form is an assembly intermediate. In any case, if this is an active intermediate, they should demonstrate this with at least a rudimentary activity assay.

The reviewer is correct that this is our hypothesis. We have added a section to the Discussion (Discussion – Protein sample) to explain the reasons why we believe our sample is CI* rather than a degradation product (see above). Our revised manuscript includes a new figure (Figure 1—figure supplement 2) showing CI*’s NADH oxidoreductase capabilities with in-gel activity and spectroscopic assays. This is in line with previous reports of CI* in-gel activity, as discussed above.

As pointed out by the reviewer, and as discussed above, we still have not tested the proton:electron ratio of CI*. We intend to carry out this quantification in the next stage of the project, where we will carry out a detailed biochemical and enzymatic characterization of CI* in solution and in reconstituted lipid vesicles.

16) The text starts to speculate about "dynamism", which is an interesting topic worthy of a review/hypothesis paper, but the data does not really provide very strong arguments in favor of such flexibility. If the authors want to make the case that their structure does indicate such flexibility, they will need to bolster their analysis.On the other hand, there are some interesting observations in the data, especially the CA subunit, that could provide the bases of a more powerful discussion.

We take the reviewer’s point. We have re-written and shortened our main discussion to focus on the structure, minimizing the speculation on CI*’s potential roles, and expanding on the findings for the CA domain, as described above.

Reviewer #2:[…]Specific comments:1) The authors did not provide any evidence on the CI* to be functional. The idea of the regulation of the H^+^/e^-^ ratio is attractive but the discussion sounds rather speculative and redundant (the last four paragraphs).

Our revised manuscript includes a new figure (Figure 1—figure supplement 2) showing CI*’s NADH oxidoreductase capabilities with standard in-gel activity and spectroscopic assays. While we intend to carry out a deeper biochemical and enzymatic characterization of CI* both in solution and in reconstituted vesicles, we believe this is outside the scope of the current manuscript, especially given the practical limitations imposed by the pandemic at this time. We have re-written and shortened our discussion to minimize the speculation on CI*’s potentially altered proton pumping. We also provide supplementary discussion in Appendix with further details of the bioenergetic arguments.

2) Plant mitochondria function in consuming excessive reducing power generated in chloroplasts and the proposed idea is interesting. But the CI* complexes were isolated from etiolated seedlings. Is it possible to detect the CI* complexes in green tissues?

Although obtaining CI* from green tissues is beyond the scope of our manuscript, several other groups have detected CI* in green and non-etiolated tissues (both seedlings and mature leaves) of *A. thaliana* and *N. sylvestris* (Pineau et al., 2008, Meyer et al., 2011, Schertl et al., 2012, Schimmeyer et al., 2016, Senkler et al., 2017, Ligas, et al., 2019). We have clarified this in our revised main text and Discussion.

A possible reason for the apparently higher levels of CI* in our preparations of etiolated *V. radiata* compared to the previous reports with non-etiolated samples may be due to the seedlings’ consumption of their oil seeds during development and lack of other energy sources. We have expanded our discussion of these issues in an Appendix to ensure clarity and brevity in the main text.

3) They did not show the activity of the CI* complex. Is there any genetic evidence supporting that the complex I is functional in the absence of PD?

There is genetic evidence for this in other organisms. Knockout of *Yarrowia lipolytica* (yeast) mitochondrial CI subunit nb8m (*nb8m∆*) has been reported to prevent assembly of its CI distal-pump domain. The *nb8m∆* CI subcomplex, which is analogous to CI*, is functional and pumps protons at half the H^+^:e^-^ ratio of fully assembled CI (i.e. 2H^+^:2e^-^ as opposed to the standard 4H^+^:2e^-^ of intact CI). Additionally, deletion or truncation of *E. coli*’s NuoL (proximal-pump domain homologue of NU5M, which bridges the proximal and distal pumps) results in a functional CI that pumps protons at a 2H^+^:2e^-^ ratio (Steimle et al., 2011). We have clarified this in the revised discussion and added the *E. coli* reference, as per reviewer 3’s recommendation. As described above, we have also included an additional figure showing CI* NADH-oxidoreductase activity (Figure 1—figure supplement 2).

The study of nad4 or nad5 (P_D_ pumps) deletion mutants is challenging given that these genes are mitochondrially encoded. In plants, to our knowledge there is no direct genetic evidence supporting or refuting the hypothesis that CI* is a functional NADH-quinone dehydrogenase. However, it is known that complete absence of CI is not lethal to plants (Kuhn et al., 2015, Colas des Francs-Small and Small, 2014). Moreover, some “surrogate mutants” in nad4 or nad5 expression do not show macroscopic phenotypes (Colas des Francs-Small and Small, 2014 and references therein).

A paper studying the differences between *Arabidopsis* plants completely lacking CI due to mutation of the core flavoprotein subunit NDUFV1 (matrix arm) and those with reduced CI activity due to the mutation of the accessory subunit NDUFS4 (matrix arm) claims that CI subcomplexes are non-functional (Kuhn et al., 2015). This is based on their indirect evidence that *ndufv1* and *ndufs4* mutants both accumulate the same CI assembly subcomplexes. The argument is that, given that the CI subcomplexes in the severe *ndufv1* mutants are not able to rescue at least some CI activity, this implies that CI subcomplexes in general (Kuhn et al., 2015), and CI* in particular (Schimmeyer et al., 2016), are non-functional. We find this unconvincing. Firstly, there is no characterization of the CI subcomplexes of either mutant beyond one unclear Western blot. Secondly, more importantly, the subcomplexes of CI (including CI*) of the *ndufv1* mutants would also be lacking NDUFV1. Thus, these subcomplexes are not the same as wild-type subcomplexes and, given the lack of NDUFV1, would also not be expected to be functional in this mutant.

On the other hand, several studies have found that plant surrogate mutants with impaired expression of nad4 and nad5 (the P_D_ pumps) accumulate CI* (Karpova and Newton, 1999, Pineau et al., 2008, Haili et al., 2013). A study of maize (Karpova and Newton, 1999) found that heteroplasmic nad4 mutants accumulate a “smaller, faster migrating, partially assembled” CI subcomplex that has NADH-dehydrogenase activity in the standard in-gel activity assay. Interestingly, the more severe the mutant phenotype, the higher the levels of the CI subcomplex. Given the timing of this study, however, it was not determined whether this subcomplex actually is CI*, as the CI* designation had yet to be defined. A study of *Arabidopsis* mutants of *mtsf1* – an mRNA-binding protein essential for the stabilization and 3′-end processing of mitochondrial *nad4* mRNA – found that these plants accumulate CI*, which is active in in-gel assays (Haili et al., 2013). Moreover, these mutants showed significantly higher oxygen consumption and lower carbon dioxide assimilation than wild-type. Although not understood, these observations were deemed “likely to stem from both a higher commitment of electrons to the AOX pathway and inefficient electron donation, thereby leading to an uncoupling effect”. Although speculative, these observations are also consistent with the activity of CI* pumping with a 2H^+^:2e^-^ ratio.

4) I am unsure why they focused on the assembly intermediate before clarifying the structure of the fully assembled complex I.

As discussed in the manuscript, we believe the structure of CI* already offers a wealth of information about the differences between plant/mammalian/yeast/bacteria CI, its assembly pathway and additional potential biological implications.

5) P5, L18. NUM2?

Thank you. Corrected to NU2M.

Reviewer #3:[…]I am, in general, enthusiastic about this manuscript and consider it within the remit for publication within eLife. It is well written, and the description of the structure especially seems very sound. I do, however, have questions for the authors concerning the methodology, and in particular, the Discussion.i) The enzyme, as isolated from etiolated mung bean by detergent treatment, is lacking two major (proton-translocating) subunits from the membrane domain (the “PD” domain) and is described throughout the manuscript as CI*, as an intermediate, assembly form of Complex I. This is, in principle, acceptable, as this CI* intermediate has been documented by others elsewhere. However, it does raise the question whether this is an artefact of isolation (i.e. JBC 278: 43114 , 2003). Did the authors observe any intact CI in their preparation, or evidence for a solubilised NuoL/M subcomplex? It also remains to be seen if CI* is a peculiarity to etiolated tissue, and perhaps the authors can expand on this.

We agree that this is an important point, which we should have emphasized in our manuscript. The arguments that support that CI* is an assembly intermediate and not a degradation product or a peculiarity of etiolated tissue are below. We have included these points in a new section in the Discussion of our revised manuscript (Discussion – Protein sample) and have added a new supplementary figure showing details of CI*’s purification and enzymatic activity (Figure 1—figure supplement 2).

Please also see response to Reviewer #1 point 2.

ii) The enzymatic activity of the CI* preparation was not determined. The authors suggest, that it functions as a 2H^+^/2e pump, by analogy with mutant forms of the fungal enzyme. This in itself is a reasonable suggestion, as this stoichiometry has also been observed with similar mutants of *E. coli* Complex I (Biochemistry 50: 3386, 2011), however, it is a pity that this was not investigated here as a discussion of the bioenergetics of CI* form a central part of the manuscript.

The revised manuscript includes a new figure (Figure 1—figure supplement 2) showing CI* that is enzymatically active in standard in-gel and spectroscopic activity assays. This is consistent with CI*’s previously reported in-gel activity, as described above. We thank the reviewer for the additional reference, which we now include in our revised Discussion. We have also added an Appendix with details of our bioenergetic arguments for the potential roles of CI*.

iii) The authors write that "Moreover, in photosynthetic tissue, conditions of intense light may lead to an over-production of NADH". I think this statement needs qualification for the benefit of non-expert readers. Obviously, the primary reductant produced by photosynthetic linear electron flow is NADPH and plants have many mechanisms to deal with sudden increases in light intensity (induction of NPQ, photosynthetic control at the level of cytochrome b6f etc.). Now, it is true that “excess” reducing equivalents are transferred from the chloroplast via the malate shunt, but this is a relatively low flux pathway, and so I assume that the authors are referring to NADH produced by the photorespiratory pathway via glycine decarboxylase activity. It is reasonable to assume that this may cause relatively large fluctuations in intramitochondrial [NADH]. (This also, obviously, will not be a factor in non-photosynthetic/etiolated tissue).Related to point ii) above, the authors write "If the NADH/NAD+ ratio dropped in photosynthetic cells due to decreased light availability, this would lead to a decrease in free energy available to CI for H^+^ pumping. Full-length CI must pump 4 protons per 2 NADH electrons, it cannot operate in a graduated manner. Consequently, at a decreased NADH/NAD+ ratio, with lower free energy available, full-length CI would initially not be able to pump any protons at all. This would result in a reduction in the mitochondrial membrane potential, decreasing the proton-motive force and lowering ATP generation."The thinking seems muddled here. Firstly, Complex I can be assumed to be operating in equilibrium (or near equilibrium) with the PMF, and so a “variable” H^+^/e pumping stoichiometry afforded by CI*/CI provides no benefit. The authors can demonstrate this for themselves through elementary thermodynamic principles if they seek reassurance (i.e. the equilibrium relationships (H^+^/2e)*PMF = 2ΔE_h_ and also K = 10^((ΔE_h_-(PMF*(H^+^/2e))/30) etc) – in fact, I would encourage this as a supplementary figure. I doubt that there will be circumstances under which a supply of reducing equivalents to the plant mitochondrial respiratory chain will become limiting, particularly given the menagerie of electron donors that supply it. I can think of one instance in which 2H^+^/2e translocating CI* would provide a benefit over a 4H^+^/2e counterpart; it would be effectively irreversible at high (>200 mV) PMF, which may limit ROS production due to reverse ET (clearly the non-protonmotive NDA enzymes can be considered irreversible under all circumstances).

We thank the reviewer for this comment, which drove the creation of our Appendix, where we examine the bioenergetic arguments in more detail. Please see this Appendix and our response to reviewer #1’s points 12 and 13.

iv) The suggestion that the γCA subunit may be acting as a bicarbonate sensor by conformational change transmitted to the membrane domain seems very speculative.

Indeed, our structure does not provide evidence for any potential influence of CA conformational changes on CI’s NADH-ubiquinone oxidoreduction, and this currently remains speculative. However, our structure shows that the interface between the CA domain and the membrane arm is extensive, with protein:protein interactions of the CA domain with several of the membrane-arm subunits. Using the PDBePISA protein interface tool, we estimate that the interface between the CA domain and the membrane arm is ~3,740 A^2^. We have added the estimated size of the interface to the main text, together with an additional supplementary table (Table 5). Given this large surface and the number of protein:protein interactions, it is feasible that a conformational change in the CA domain could propagate to the N or Q modules to affect NADH-ubiquinone oxidoreduction.

However, given the speculative nature of the hypothesis and the reviewers’ comments, we have removed it from the Discussion.

v) Typo – Figure 1—figure supplement 3 – “accesory” in many places.

Thank you. Corrected.

References:

Colas des Francs-Small, C and Small, I. Surrogate mutants for studying mitochondrially encoded functions, Biochimie, Volume 100, 2014, Pages 234-242, https://doi.org/10.1016/j.biochi.2013.08.019.

Heddy Soufari, Camila Parrot, Lauriane Kuhn, Florent Waltz, Yaser Hashem

bioRxiv 2020.02.21.959148; doi: https://doi.org/10.1101/2020.02.21.959148

Iñiguez C, Capó-Bauçà S, Niinemets Ü, Stoll H, Aguiló-Nicolau P, Galmés J. Evolutionary trends in RuBisCO kinetics and their co-evolution with CO_2_ concentrating mechanisms. Plant J. 2020;101(4):897‐918. doi:10.1111/tpj.14643

Klanchui et al. (2017). Exploring Components of the CO_2_-Concentrating Mechanism in Alkaliphilic Cyanobacteria Through Genome-Based Analysis. Comput Struct Biotechnol J. 15:340-350. doi: 10.1016/j.csbj.2017.05.001

Kühn K, Obata T, Feher K, Bock R, Fernie AR, Meyer EH. Complete Mitochondrial Complex I Deficiency Induces an Up-Regulation of Respiratory Fluxes That Is Abolished by Traces of Functional Complex I. Plant Physiol. 2015;168(4):1537‐1549. doi:10.1104/pp.15.00589

Long et al. (2016). Cyanobacterial CO_2_-concentrating mechanism components: function and prospects for plant metabolic engineering. Curr Opin Plant Biol.31:1-8. doi: 10.1016/j.pbi.2016.03.002.

Supuran and Capasso (2017). An Overview of the Bacterial Carbonic Anhydrases. Metabolites 7. doi: 10.3390/metabo7040056

Schuller, J. M. et al. (2020). Redox-coupled proton pumping drives carbon concentration in the photosynthetic complex I. Nat. Commun. 11, 494–7.